# FLASHMASK: EFFICIENT AND RICH MASK EXTENSION OF FLASHATTENTION

Guoxia Wang[*], Jinle Zeng[*], Xiyuan Xiao, Siming Wu,
Jiabin Yang, Lujing Zheng, Zeyu Chen, Jiang Bian,
Dianhai Yu[†], Haifeng Wang

Baidu Inc.

{wangguoxia, zengjinle, yudianhai}@baidu.com

## ABSTRACT

The computational and memory demands of vanilla attention scale quadratically with the sequence length $N$, posing significant challenges for processing long sequences in Transformer models. FlashAttention alleviates these challenges by eliminating the $O(N^2)$ memory dependency and reducing attention latency through IO-aware memory optimizations. However, its native support for certain attention mask types is limited, and it does not inherently accommodate more complex masking requirements. Previous approaches resort to using dense masks with $O(N^2)$ memory complexity, leading to inefficiencies. In this paper, we propose FLASHMASK, an extension of FlashAttention that introduces a column-wise sparse representation of attention masks. This approach efficiently represents a wide range of mask types and facilitates the development of optimized kernel implementations. By adopting this novel representation, FLASHMASK achieves linear memory complexity $O(N)$, making it suitable for modeling long-context sequences. Moreover, this representation enables kernel optimizations that eliminate unnecessary computations by leveraging sparsity in the attention mask, without sacrificing computational accuracy, resulting in higher computational efficiency. We evaluate FLASHMASK's performance in fine-tuning and alignment training of LLMs such as SFT, LoRA, DPO, and RM. FLASHMASK achieves significant throughput improvements, with end-to-end speedups ranging from 1.65x to 3.22x compared to existing FlashAttention dense method. Additionally, our kernel-level comparisons demonstrate that FLASHMASK surpasses the latest counterpart, FlexAttention, by 12.1% to 60.7% in terms of kernel TFLOPs/s, achieving 37.8% to 62.3% of the theoretical maximum FLOPs/s on the A100 GPU. The code is open-sourced on PaddlePaddle[1] and integrated into PaddleNLP[2], supporting models with over 100 billion parameters for contexts extending up to 128K tokens.

## 1 INTRODUCTION

The Transformer architecture Vaswani et al. (2017) has become a foundational model in a wide range of tasks across natural language processing (NLP), computer vision (CV), and multimodal applications. Central to its effectiveness is the attention mechanism, which enables the model to focus on relevant parts of the input data. In the vanilla attention mechanism, the attention weights are computed as a scaled dot-product between query and key vectors, as shown in Equation (1). To implement complex logic, the mask $M$ can be added to the $QK^T$ term before applying the softmax function, controlling token visibility by setting certain elements to $-\infty$.

---

[*]Equal contribution.

[†]Corresponding author.

[1]https://github.com/PaddlePaddle/Paddle

[2]https://github.com/PaddlePaddle/PaddleNLP

$$\text{Attention}(Q, K, V, M) = \text{Softmax}\left(\frac{QK^T}{\sqrt{d_k}} + M\right) V \tag{1}$$

In NLP tasks, large language model (LLM) training can generally be categorized into two main stages: PreTraining and PostTraining. During PreTraining, different masking strategies are employed to guide the model's learning process. For instance, GPT-style Radford (2018) models use unidirectional causal masking, as illustrated in Figure 1 (a)(1), while T5-style Raffel et al. (2020) models utilize a combination of unidirectional and bidirectional masking, as shown in Figure 1 (a)(9). PostTraining, which typically includes Supervised Fine-Tuning (SFT) Iyer et al. (2022); Chung et al. (2024); Hu et al. (2021); Liu et al. (2024a); Chen et al. (2023), Direct Preference Optimization (DPO) Rafailov et al. (2024); Dong et al. (2023); Liu et al. (2023); Ethayarajh et al. (2024); Liu et al. (2024b), and Reward Model (RM) training within Reinforcement Learning from Human Feedback (RLHF) Schulman et al. (2017); Shao et al. (2024), also employs a variety of masking techniques depending on the specific task, as depicted in Figure 1 (a)(3) and (5).

The increasing complexity and diversity of masking strategies pose significant challenges to the efficiency of the attention mechanism. Specifically, the vanilla attention mechanism suffers from a quadratic increase in computational and memory demands, denoted as $O(N^2)$, where $N$ represents the sequence length. As models scale to longer sequences, ranging from 128K to 1M tokens in advanced systems like GPT-4 Achiam et al. (2023), Claude Anthropic (2024), and Gemini Reid et al. (2024), these quadratic dependencies become prohibitive, necessitating more efficient computational approaches. The memory load for masked attention computations also grows quadratically, further exacerbating the challenge of managing various mask configurations across different tasks.

Recent efforts such as Memory Efficient Attention (MEA) Rabe & Staats (2021) and FlashAttention Dao et al. (2022); Dao (2023) have made strides in addressing these issues by reducing memory overhead and attention latency. FlashAttention, in particular, eliminates the $O(N^2)$ memory dependency and reduces attention latency through IO-aware memory read/write optimizations. However, while FlashAttention natively supports certain mask types without additional memory overhead, its support for more complex masking requirements remains limited.

In this paper, we introduce **FLASHMASK**, an extension of FlashAttention that leverages a novel column-wise representation of attention masks. This approach allows for the efficient handling of a broader range of mask types without compromising computational accuracy. FLASHMASK achieves linear memory complexity while enabling kernel optimizations that reduce unnecessary computations, resulting in significant computational speedups and enhanced training efficiency.

Our contributions are threefold:

1. We introduce a novel column-wise sparse mask representation that efficiently accommodates a broader range of mask types, enabling more flexible attention mechanisms.

2. We extend FlashAttention's masking capabilities by integrating optimized kernel implementations, ensuring high computational efficiency without sacrificing computational accuracy.

3. We demonstrate the effectiveness of FLASHMASK across various attention mask types and models, underscoring its versatility and robustness in large-scale LLM training. FLASH-MASK notably reduces both computational and memory overheads, significantly enhancing its suitability for long-context modeling.

## 2 BACKGROUND

### 2.1 ATTENTION MASK TYPES

Transformer-based models have demonstrated exceptional versatility across a variety of tasks, each benefiting from different attention mask types, as shown in Figure 1. *Causal Mask* is predominantly used in autoregressive models to predict the next token in a sequence, ensuring that each token only attends to previous tokens and avoids information leakage from future tokens Vaswani et al. (2017). *Sliding Window Mask* captures local context by allowing tokens to attend to a fixed-size window of neighboring tokens, balancing computational efficiency with the ability to capture local dependencies Beltagy et al. (2020). *Causal Document Mask*, employed in methods like efficient sequence packing and in-batch/in-tokens techniques, accelerates large language models without

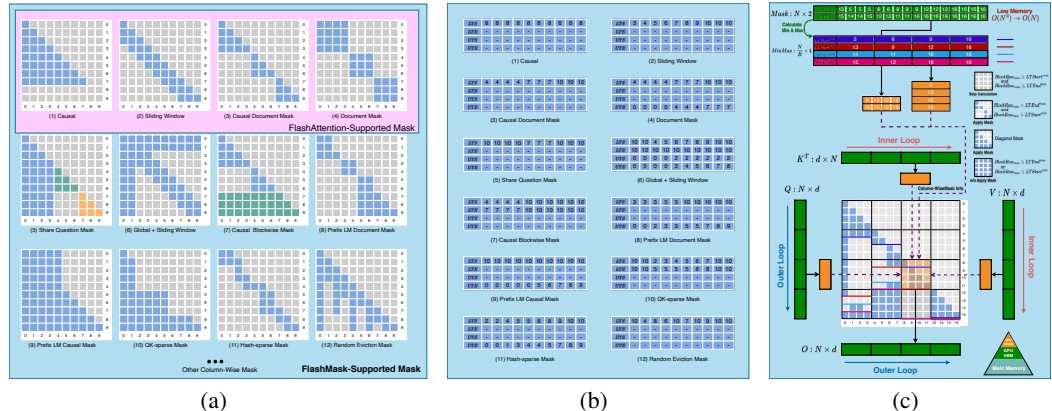

Figure 1: Overview of FLASHMASK. (a) Types of Masks Supported by FLASHMASK, (b) Column-Wise Sparse Representation in FLASHMASK, (c) Efficient Implementation of FLASHMASK.

performance degradation by ensuring tokens attend only to previous tokens within the same document Krell et al. (2021); Iyer et al. (2022); Dubey et al. (2024). *Document Mask*, or bi-directional attention, permits tokens to attend to all other tokens within the same document, facilitating context learning from both directions and is widely used in models like BERT and vision transformers like NaViT Devlin et al. (2018); Dehghani et al. (2024).

*Shared Question Mask* is utilized in Reward Models (RM) and Direct Preference Optimization (DPO) models, allowing multiple answers to share a single question, thus eliminating redundant computations and speeding up training Ouyang et al. (2022). The *Global + Sliding Window Mask* combines global attention with sliding window attention, where global tokens attend to all tokens while others use a sliding window mask, effectively handling tasks requiring both global context and local details Zaheer et al. (2020).

*Causal BlockWise Mask*, primarily used in in-context learning, divides sequences into blocks, where demonstrations only attend to nearby examples within small blocks, while the test example can attend to all demonstrations, allowing the study of model performance improvements in long-context tasks Bertsch et al. (2024). *Prefix LM Causal Mask* is tailored for language modeling tasks, allowing a prefix to attend to all tokens to generate coherent text based on the prefix Raffel et al. (2020). *Prefix Document Mask* extends this concept to multiple documents, where a prefix in each document attends to all tokens within that document but not across documents.

*QK-Sparse Mask* optimizes self-attention by sparsifying query-key pairs, reducing computational load while maintaining performance, which is particularly beneficial for large-scale models Kitaev et al. (2020). *Hash-Sparse Mask* employs locality-sensitive hashing to partition sequences into smaller chunks, enabling efficient sparse attention for long sequences Kitaev et al. (2020). Lastly, *Random Eviction Mask* introduces randomness by randomly masking out tokens during training, aiding in generalization and simulating Key-Value (KV) cache eviction processes to handle long sequences without memory overflow Chen et al. (2024).

## 2.2 ATTENTION MASK SUPPORTED

Attention mechanisms are fundamental to transformer-based models, with various mask types enabling different attention patterns. The vanilla attention mechanism, as shown in Equation 2, supports arbitrary mask types through a dense mask matrix:

$$S = \frac{QK^\top}{\sqrt{d_k}} \in \mathbb{R}^{N \times N}, \quad P = \text{Softmax}(S + M) \in \mathbb{R}^{N \times N}, \quad O = PV \in \mathbb{R}^{N \times d}, \quad (2)$$

where $Q$, $K$, $V \in \mathbb{R}^{N \times d}$ are input sequences, $M \in \mathbb{R}^{N \times N}$ is the attention mask, $N$ is the sequence length, and $d$ is the head dimension. The mask $M$ modulates token visibility through element-wise addition with $S$. While this approach supports arbitrary mask types, it incurs a memory complexity of $O(N^2)$, limiting its scalability for long sequences.

FlashAttention Dao et al. (2022); Dao (2023) addresses this limitation through IO-aware read/write operations and tiling techniques, eliminating the need for the intermediate $S \in \mathbb{R}^{N \times N}$ and explicit mask $M$. However, FlashAttention only supports predetermined mask patterns within its kernel, such as causal, sliding window, causal document, and document masks, as shown in Figure 1.

xFormers Lefaudeux et al. (2022) extends FlashAttention's capabilities, offering support for masks with diagonal offsets. It represents document masks using cumulative sequence lengths, achieving a memory complexity of $O(N)$.

FlexAttention He et al. (2024) introduces a more flexible mask description method based on deep learning compiler techniques. By combining block masks with expression-based descriptions, it can support arbitrary mask types. While this approach significantly reduces memory overhead through block-based processing, its memory complexity remains $O(\frac{N^2}{B_r B_c})$.

Our proposed method, FLASHMASK, extends FlashAttention's mask support capabilities. It introduces a flexible, column-wise sparse mask representation that covers the majority of mainstream Transformer modeling requirements. As illustrated in Figure 1(b), FLASHMASK expresses which intervals need to be masked on a per-column basis, achieving a memory complexity of $O(N)$. This approach bridges the gap between mask flexibility and computational efficiency, offering a more versatile solution for attention mechanisms in large-scale transformer models.

### 2.3 ATTENTION OPTIMIZATION TECHNIQUES

The attention mechanism, as formulated in Equation 2, presents significant computational and memory challenges, particularly in the computation of $QK^T$. As the sequence length $N$ increases, the resultant attention scores matrix grows quadratically, leading to a complexity of $O(N^2)$. To address this scalability issue, researchers have proposed various optimization techniques, focusing on both memory efficiency and computational speed.

Memory Efficient Attention (MEA) Rabe & Staats (2021) marks a notable advancement in model training optimizations. By leveraging Online Softmax Milakov & Gimelshein (2018) and chunking techniques, MEA reduces memory requirements from $O(N^2)$ to $O(\sqrt{N})$, enabling the use of larger models or extended sequence lengths within existing hardware constraints. Building upon this foundation, FlashAttention Dao et al. (2022); Dao (2023) focuses on reducing attention latency through IO-aware memory read/write optimizations. Utilizing tiling techniques during computation, FlashAttention achieves a memory overhead of $O(N)$, proving particularly effective in tasks without custom masking requirements. Furthermore, FlashAttention extends to Block-Sparse FlashAttention, introducing a two-dimensional block mask matrix representation to indicate masked tiling blocks. This innovation allows for the skipping of computations for masked blocks, thereby accelerating the process.

For scenarios requiring specific attention masks, several tailored solutions have emerged. Sparse Causal Flash Attention (SCFA) Pagliardini et al. (2023) extends FlashAttention to optimize QK-Sparse and Hash-Sparse scenarios in causal attention structures. SCFA employs indices of queries and keys in the original uncompressed tensors to describe masks, enabling the omission of computations for masked blocks and enhancing computational efficiency. FlexAttention He et al. (2024) leverages compiler techniques to simplify mask attention implementations, exploiting sparsity in the attention mask to skip certain masked blocks and achieve improved speed. However, there remains room for optimization, particularly for complex masking patterns.

Our proposed method, FLASHMASK, builds upon these advancements to support customized complex attention masks. FLASHMASK reduces memory complexity from $O(N^2)$ to $O(N)$ while leveraging sparsity in the attention mask to skip masked blocks. Through rigorous engineering optimizations, FLASHMASK achieves superior computational speed compared to FlexAttention, particularly in tasks with complex masking requirements. By synthesizing the strengths of existing approaches with novel optimization techniques, FLASHMASK represents a significant advancement in attention mechanism efficiency, addressing both memory constraints and computational speed in large-scale transformers.

## 3 OBSERVATION

In the current paradigm of training Transformer-based models, attention mechanisms can be broadly categorized based on their causality, as introduced in Section 2.1 and illustrated in Figure 1(a). These representations encompass the majority of mask types encountered in training scenarios. Consider the attention score matrix $S$, where each element $S_{ij}$ represents the attention of the $i$-th query token to the $j$-th key token. From the perspective of the key tokens, we observe a critical pattern in how query tokens attend to each key token.

Our key observation is that the inability of query tokens to attend to certain key tokens exhibits a continuous nature. This continuity allows us to transform the two-dimensional dense mask $M$ into a more compact, one-dimensional representation using row index intervals, as depicted in Figure 1(b). Formally, we can express this transformation as:

$$M_j = [start_j, end_j), \quad \forall j \in \{1, \dots, N\} \tag{3}$$

where $M_j$ represents the interval of row indices that are masked for the $j$-th key token, and $N$ is the sequence length.

While this column-wise, one-dimensional interval representation may not capture arbitrary mask patterns, it effectively covers the predominant mask types encountered in practice. Moreover, this representation offers a significant advantage: it facilitates a straightforward conversion to masked blocks in tiling-based computations. This conversion enables the elimination of unnecessary calculations, thereby enhancing the computational efficiency of the attention kernel.

This concept of interval representation can be generalized to other forms of continuous intervals. For instance, by transposing the attention matrix, we can obtain a row-wise representation using column index intervals. The flexibility of this approach allows for efficient handling of various attention patterns while maintaining a compact representation that is conducive to optimized computation.

## 4 FLASHMASK: ALGORITHM AND ANALYSIS

In this section, we introduce the novel column-wise mask representation of FLASHMASK, extend FlashAttention to support complex mask patterns, and provide a comprehensive complexity analysis of our approach.

### 4.1 COLUMN-WISE MASK REPRESENTATION

To efficiently handle complex mask patterns in both causal and bidirectional attention scenarios, we propose a novel column-wise sparse representation for FLASHMASK. The attention score matrix is partitioned into lower-left and upper-right triangular sections relative to the diagonal. FLASHMASK expresses the mask using four one-dimensional vectors:

- **$LTS$**: **L**ower **T**riangular **S**tart - the starting row of the mask in the lower-left triangle.
- **$LTE$**: **L**ower **T**riangular **E**nd - the ending row of the mask in the lower-left triangle.
- **$UTS$**: **U**pper **T**riangular **S**tart - the starting row of the mask in the upper-right triangle.
- **$UTE$**: **U**pper **T**riangular **E**nd - the ending row of the mask in the upper-right triangle.

The indices of rows to be masked in the lower triangular section are given by $[LTS, LTE)$, and in the upper triangular section by $[UTS, UTE)$. Specifically, each column is described by two mask intervals. For the $j$-th token, tokens within the intervals $[LTS_j, LTE_j) \cup [UTS_j, UTE_j)$ cannot attend to it. For example, as illustrated in Figure 1(b)(6), for the fifth column, $[LTS_5, LTE_5) \cup [UTS_5, UTE_5) = [7, 10) \cup [2, 4)$ indicates that rows 2 to 4 and 7 to 9 are masked.

This representation offers several advantages:

1. **Compactness**: It reduces a dense 2D mask to a more efficient 1D representation.
2. **Flexibility**: It can capture a wide range of practical mask patterns, including causal, bidirectional, and more complex attention mechanisms.
3. **Computational Efficiency**: It facilitates easy conversion to masked blocks in tiling-based computations, enabling the elimination of unnecessary calculations.

## 4.2 EXTENDING FLASHATTENTION FOR COMPLEX MASKS

We integrate the column-wise mask representation of FLASHMASK into the FlashAttention-2 algorithm, extending its mask support capabilities. The high-performance kernel implementation of FLASHMASK consists of two key steps:

**Preprocessing**: Given the input column-wise sparse mask vectors, we first partition them into $T_c$ blocks along the column dimension in high-bandwidth memory (HBM). For each mask vector, we compute the maximum and minimum values within each block. This results in 8 intermediate vectors: $LTStart^{max}$, $LTStart^{min}$, $LTEnd^{max}$, $LTEnd^{min}$, $UTStart^{max}$, $UTStart^{min}$, $UTEnd^{max}$, and $UTEnd^{min}$, each of size $T_c$.

**Real-time Block Skip Computation**: Using these min-max vectors, we can classify each tiling block of attention score matrix into three categories during kernel computation. The block mask type $T_{block}$ is determined as follows:

$$T_{block} = \begin{cases} \text{Fully masked,} & \text{if } BlockRow_{min} \geq Start^{max} \text{ and } BlockRow_{max} \leq End^{min} \\ \text{Partially masked,} & \text{elif } BlockRow_{min} < End^{max} \text{ and } BlockRow_{max} > Start^{min} \\ \text{Unmasked,} & \text{otherwise} \end{cases} \quad (4)$$

This classification allows us to skip fully masked blocks, reduce computation for unmasked blocks, and apply element-wise masking only for partially masked blocks. Figure 1(c) illustrates the entire kernel computation process for a causal scenario with $LTS$ and $LTE$ in the lower-left triangle. Algorithm 1 details the forward computation process of FLASHMASK extended from FlashAttention-2, with blue-shaded parts indicating FLASHMASK computations.

For the backward pass, FLASHMASK's column-sparse representation is particularly advantageous. The computations of $dK$ and $dV$ are column-parallel, allowing efficient loading of maximum and minimum values into registers for extensive data reuse during block computations, as shown in Algorithm 2 in the Appendix.

## 4.3 COMPLEXITY ANALYSIS

We define block sparsity in attention mask as $\rho = \frac{\alpha}{\lceil \frac{N}{B_r} \rceil \times \lceil \frac{N}{B_c} \rceil}$, where $B_r$, $B_c$ are block sizes, and $\alpha$ is the number of completely masked blocks.

**Space Complexity:** The dense mask requires $O(N^2)$ space, while FLASHMASK uses $O(N)$ space for $LTS, LTE, UTS, UTE \in \mathbb{R}^N$ and 8 precomputed min-max vectors $\in \mathbb{R}^{\lceil \frac{N}{B_c} \rceil}$. This significant reduction in memory usage enables training on longer sequences.

**Memory Access Complexity:** The dense mask requires $O(N^2)$ memory accesses on HBM. FLASHMASK reads the **LTS, LTE, UTS, UTE** $\in \mathbb{R}^N$ vectors from HBM as shown in lines 16 and 19 of Algorithm 1, with each $\mathbf{Q}_i$ reading the entire **LTS, LTE, UTS, UTE**, totaling $4 \times T_r \times N$ memory accesses. This reduces the memory access to approximately $\frac{N^2}{4 \times T_r \times N} = \frac{B_r}{4}$, significantly boosting performance. Furthermore, FLASHMASK's compact representation allows preloading of mask vectors into SRAM, further enhancing memory access efficiency.

**Computational Complexity:** While the standard attention computation has a complexity of $O(N^2)$, FLASHMASK leverages sparsity in the attention mask to reduce it to $O((1-\rho)T_r T_c)$ by skipping entirely masked blocks.

These improvements in space, memory access, and computational complexities contribute to FLASHMASK's superior performance and efficiency in handling complex attention patterns.

## 4.4 CORRECTNESS ANALYSIS

As shown in Equation 2, the computation of the attention matrix $P = \text{Softmax}(S + M)$ involves augmenting the attention scores $S$ with a mask $M$, where the masked elements are set to $-\infty$. This operation ensures that the softmax outputs at these masked positions are zero, effectively omitting them from attention. Consequently, if an entire block is fully masked, the resulting output for that block will be all zeros. FLASHMASK exploits sparsity in the attention mask by skipping computations involving these entirely masked blocks, thus reducing computational overhead without altering the

---

**Algorithm 1** FlashAttention-2 Forward Pass Extended with FLASHMASK

---

**Require:** Matrices $\mathbf{Q}, \mathbf{K}, \mathbf{V} \in \mathbb{R}^{N \times d}$ in HBM, block sizes $B_c, B_r$, vectors $\mathbf{LTS}, \mathbf{LTE}, \mathbf{UTS}, \mathbf{UTE} \in \mathbb{R}^N$.

1: Divide $\mathbf{Q}$ into $T_r = \left\lceil \frac{N}{B_r} \right\rceil$ blocks $\mathbf{Q}_1, \ldots, \mathbf{Q}_{T_r}$ of size $B_r \times d$ each, and divide $\mathbf{K}, \mathbf{V}$ in to $T_c = \left\lceil \frac{N}{B_c} \right\rceil$ blocks
   $\mathbf{K}_1, \ldots, \mathbf{K}_{T_c}$ and $\mathbf{V}_1, \ldots, \mathbf{V}_{T_c}$, of size $B_c \times d$ each.

2: Divide the output $\mathbf{O} \in \mathbb{R}^{N \times d}$ into $T_r$ blocks $\mathbf{O}_1, \ldots, \mathbf{O}_{T_r}$ of size $B_r \times d$ each, and divide the logsumexp $L$
   into $T_r$ blocks $L_1, \ldots, L_{T_r}$ of size $B_r$ each.

3: Divide $\mathbf{LTS}, \mathbf{LTE}, \mathbf{UTS}, \mathbf{UTE}$ into $T_c$ blocks $\mathbf{LTS}_1, \ldots, \mathbf{LTS}_{T_c}, \mathbf{LTE}_1, \ldots, \mathbf{LTE}_{T_c}, \mathbf{UTS}_1, \ldots, \mathbf{UTS}_{T_c},$
   $\mathbf{UTE}_1, \ldots, \mathbf{UTE}_{T_c}$, of size $B_c$ each.

4: Precompute the min and max row index $LTStart_j^{min} = min(\mathbf{LTS}_j)$, $LTStart_j^{max} = max(\mathbf{LTS}_j)$,
   $LTEnd_j^{min} = min(\mathbf{LTE}_j)$, $LTEnd_j^{max} = max(\mathbf{LTE}_j)$, $UTStart_j^{min} = min(\mathbf{UTS}_j)$, $UTStart_j^{max} = $
   $max(\mathbf{UTS}_j)$, $UTEnd_j^{min} = min(\mathbf{UTE}_j)$, $UTEnd_j^{max} = max(\mathbf{UTE}_j)$, $\forall j \in \{1, \ldots, T_c\}$, write to HBM.

5: **for** $1 \le i \le T_r$ **do**

6:     Load $\mathbf{Q}_i$ from HBM to on-chip SRAM.

7:     On chip, initialize $\mathbf{O}_i^{(0)} = (0)_{B_r \times d} \in \mathbb{R}^{B_r \times d}, \ell_i^{(0)} = (0)_{B_r} \in \mathbb{R}^{B_r}, m_i^{(0)} = (-\infty)_{B_r} \in \mathbb{R}^{B_r}$.

8:     **for** $1 \le j \le T_c$ **do**

9:        **if** $(i-1) \times B_r \ge LTStart_j^{max}$ **and** $i \times B_r \le LTEnd_j^{min}$ **then**

10:           Continue // `lower triangular skip calculation of masked block`

11:        **end if**

12:        **if** $(i-1) \times B_r \ge UTStart_j^{max}$ **and** $i \times B_r \le UTEnd_j^{min}$ **then**

13:           Continue // `upper triangular skip calculation of masked block`

14:        **end if**

15:        Load $\mathbf{K}_j, \mathbf{V}_j$ from HBM to on-chip SRAM.

16:        On chip, compute $\mathbf{S}_i^{(j)} = \mathbf{Q}_i \mathbf{K}_j^T \in \mathbb{R}^{B_r \times B_c}$.

17:        **if** $i \times B_r > LTStart_j^{min}$ **and** $(i-1) \times B_r < LTEnd_j^{max}$ **then**

18:           Load $\mathbf{LTS}_j$ and $\mathbf{LTE}_j$ from HBM to on-chip SRAM.

19:           On chip, apply mask: $\mathbf{S}_i^{(j)}[x][y] = -\infty, \forall x, y$, such that $\mathbf{LTS}_j[y] \le (i-1) \times B_r + x < \mathbf{LTE}_j[y]$

20:        **end if**

21:        **if** $i \times B_r > UTStart_j^{min}$ **and** $(i-1) \times B_r < UTEnd_j^{max}$ **then**

22:           Load $\mathbf{UTS}_j$ and $\mathbf{UTE}_j$ from HBM to on-chip SRAM.

23:           On chip, apply mask: $\mathbf{S}_i^{(j)}[x][y] = -\infty, \forall x, y$, such that $\mathbf{UTS}_j[y] \le (i-1) \times B_r + x < \mathbf{UTE}_j[y]$

24:        **end if**

25:        On chip, compute $m_i^{(j)} = max(m_i^{(j-1)}, \text{rowmax}(\mathbf{S}_i^{(j)})) \in \mathbb{R}^{B_r}, \tilde{\mathbf{P}}_i^{(j)} = exp(\mathbf{S}_i^{(j)} - m_i^{(j)}) \in$
             $\mathbb{R}^{B_r \times B_c}$ (pointwise), $\ell_i^{(j)} = e^{m_i^{j-1} - m_i^{(j)}} \ell_i^{(j-1)} + \text{rowsum}(\tilde{\mathbf{P}}_i^{(j)}) \in \mathbb{R}^{B_r}$.

26:        On chip, compute $\mathbf{O}_i^{(j)} = \text{diag}(e^{m_i^{(j-1)} - m_i^{(j)}}) \mathbf{O}_i^{(j-1)} + \tilde{\mathbf{P}}_i^{(j)} \mathbf{V}_j$.

27:     **end for**

28:     On chip, compute $\mathbf{O}_i = \text{diag}(\ell_i^{(T_c)})^{-1} \mathbf{O}_i^{(T_c)}$.

29:     On chip, compute $L_i = m_i^{(T_c)} + \log(\ell_i^{(T_c)})$.

30:     Write $\mathbf{O}_i$ to HBM as the $i$-th block of $\mathbf{O}$.

31:     Write $L_i$ to HBM as the $i$-th block of $L$.

32: **end for**

33: Return the output $\mathbf{O}$ and the logsumexp $L$.

---

outcome. Importantly, FLASHMASK maintains bit-level numerical equivalence with the computations performed using a dense mask in FlashAttention, ensuring that there is no loss in precision. This exactness is corroborated in our experimental evaluations, where we verify that the loss convergence curves from end-to-end training align precisely at the bit level (see Section 5.2).

## 5 EXPERIMENTS

In this section, we evaluate the performance of FLASHMASK through a series of experiments designed to demonstrate its end-to-end acceleration and memory efficiency under different model scales and sequence lengths, its training convergence in practical scenarios, its relationship with block sparsity in the attention mask, and its effectiveness across various attention mask patterns. All experiments were conducted on machines equipped with NVIDIA A100-SXM 80G GPUs, Intel(R) Xeon(R)

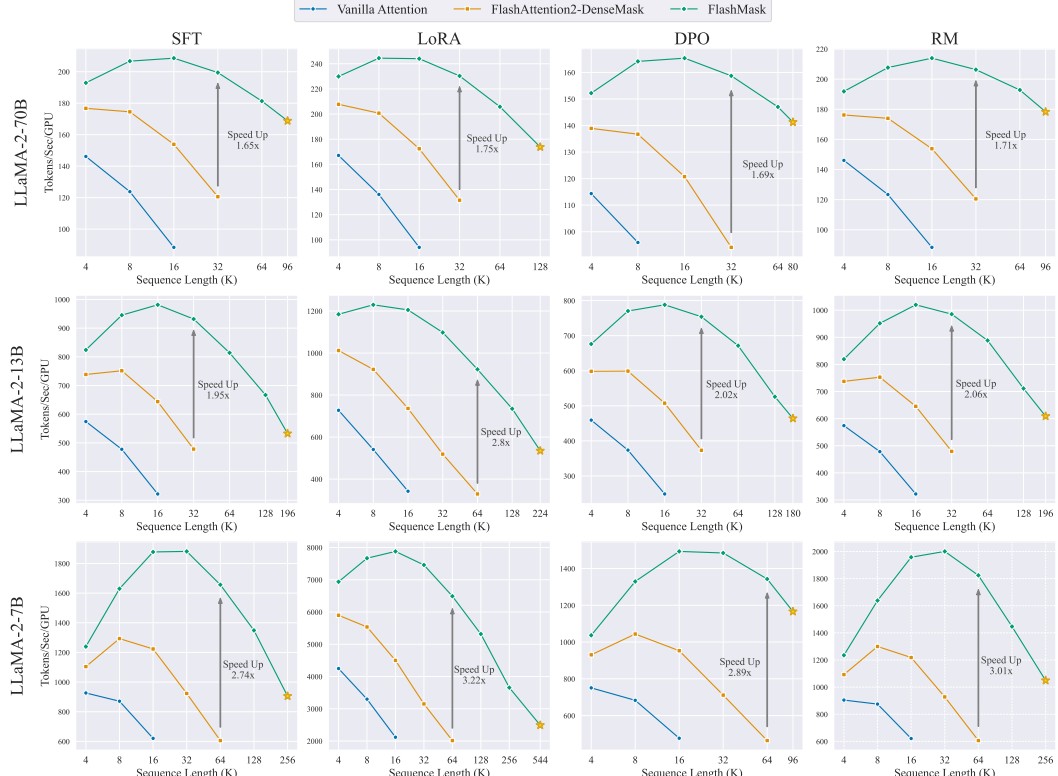

Figure 2: End-to-end training throughput was assessed across varying sequence lengths for different Llama2 model scales in four downstream training tasks: SFT, LoRA, DPO, and RM.

Platinum 8350C CPUs, CUDA 12.0, and driver version 525.125.06. Due to space limitations, detailed information about datasets and hyperparameters settings is provided in the appendix A.

## 5.1 END-TO-END TRAINING THROUGHPUT

To showcase the practical effectiveness of FLASHMASK, we evaluated the end-to-end training throughput on Llama-2 models of three scales (7B, 13B, and 70B) across four downstream tasks involving the fine-tuning and alignment training (SFT, LoRA, DPO, and RM) with varying sequence lengths. We compared FLASHMASK with two dense mask methods. The experimental results are presented in Figure 2. The results lead to two key conclusion. **Higher Throughput:** FLASHMASK achieves higher throughput compared to dense mask methods with quadratic memory complexity. Specifically, FLASHMASK attains a 1.65x to 3.22x improvement over the maximum sequence length supported by FlashAttention dense mask. This demonstrates that, in practical applications, FLASH-MASK can significantly enhance training throughput of large language models, thereby reducing training costs. **Linear Memory Overhead:** FLASHMASK's linear memory overhead enables it to support longer sequence lengths. In the Llama-2 7B LoRA training, FLASHMASK supports sequence lengths up to 544K, whereas other methods are limited to 64K. At a sequence length of 64K, the memory overhead for dense mask methods amounts to 8GB. Figure 4 (b) depicts the memory overhead curve, highlighting the efficiency of FLASHMASK in terms of memory consumption.

## 5.2 END-TO-END TRAINING CONVERGENCE VERIFICATION

The core innovation of FLASHMASK lies in introducing a column-wise sparse mask representation, which leverages sparsity in the attention mask to skip computations on fully masked blocks, thereby enhancing speed without altering the algorithm's precision. To verify that FLASHMASK does not compromise convergence accuracy, we conducted end-to-end training experiments on the Llama 3.1 Dubey et al. (2024) 8B model across four fine-tuning and alignment training tasks of LLMs.

It is important to **note that** the backward computation of **dQ** in the CUDA kernel implementation may introduce randomness due to the accumulation order (see line 27 of Algorithm 2). Therefore,

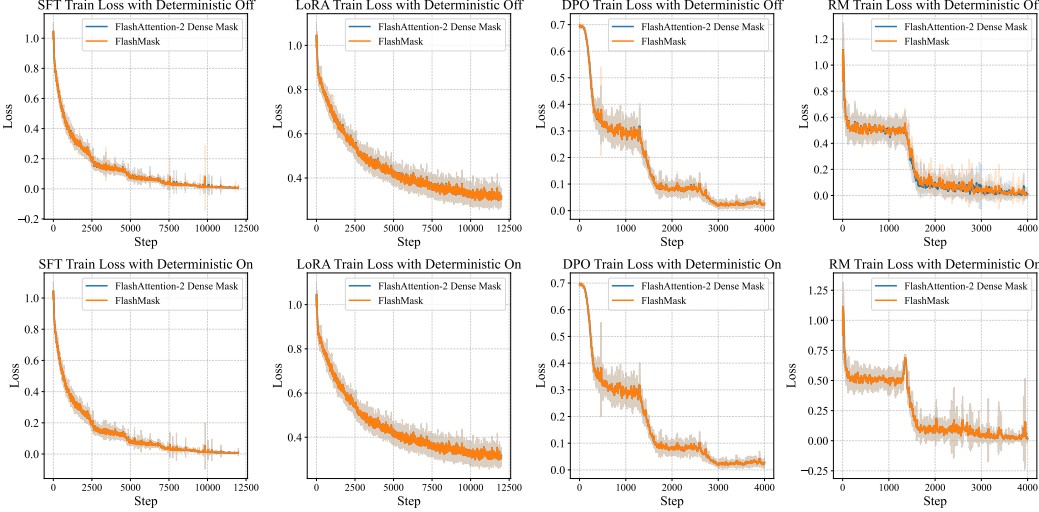

Figure 3: End-to-end training loss.

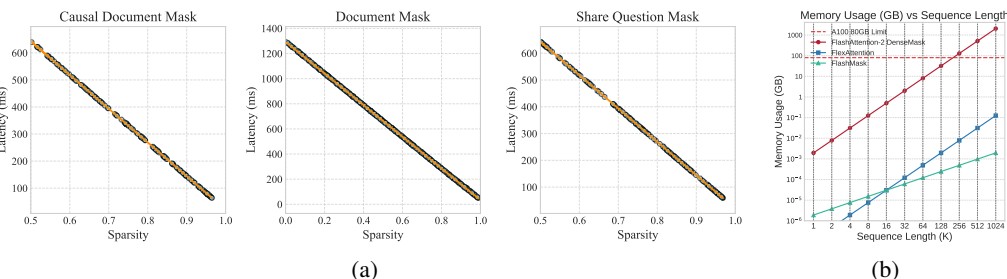

Figure 4: (a) Kernel execution latency at different sparsity levels, (b) Memory usage, Y-axis uses a base-10 logarithmic scale.

we performed convergence experiments under conditions with and without deterministic control. The loss curves are shown in Figure 3. When deterministic control is enabled, the loss curves of FLASHMASK and FlashAttention dense mask align precisely, demonstrating identical numerical behavior. When deterministic control is disabled, both methods exhibit the same loss convergence trends. These results conclusively prove that FLASHMASK is an exact algorithm that preserves convergence accuracy.

### 5.3 SPARSITY-RELATED EXPERIMENTS

FLASHMASK leverages block sparsity in the attention mask to skip computations on fully masked blocks, resulting in computational complexity proportional to $O((1 - \rho)T_r T_c)$. To verify this relationship, we performed experiments on three different mask cases under the configuration of BFloat16 data type, sequence length of 32K, head dimension of 128, and 32 heads. We sampled data with varying sparsity levels for testing. Figure 4 (a) illustrates the kernel execution latency at different sparsity levels, demonstrating a linear relationship between latency and sparsity.

### 5.4 KERNEL PERFORMANCE COMPARISON

To thoroughly evaluate the expressiveness and computational efficiency of FLASHMASK under common attention mask patterns, we conducted kernel-level comparisons with FlexAttention. The experiments were carried out across 12 different mask cases, with sequence lengths of 8K, 32K, and 128K, and head dimensions of 64 and 128, using BFloat16 data type. The total number of tokens was fixed at 128K; varying sequence lengths yielded corresponding batch sizes, and a fixed hidden size of 4096 allowed us to adjust the number of heads by changing the head dimension. Both FLASHMASK and FlexAttention exploit sparsity in the attention mask. We measured kernel speed using the TFLOPs/s metric. As shown in Figure 5, FLASHMASK outperforms FlexAttention in terms of total TFLOPs/s for both forward and backward passes across all cases, with improvements ranging

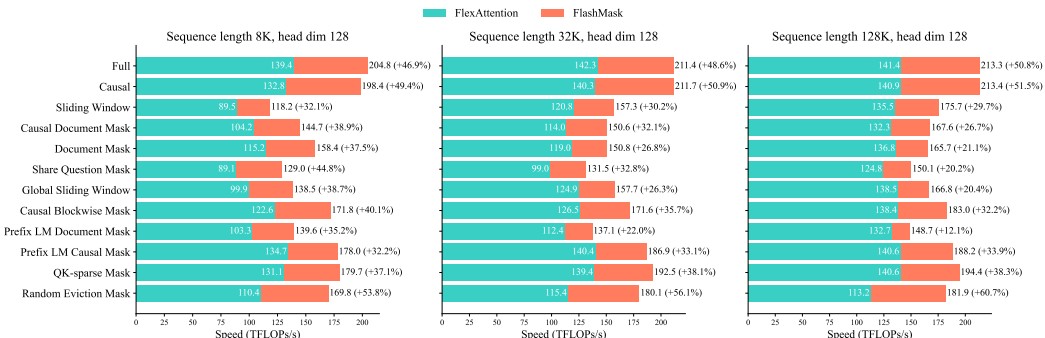

Figure 5: Kernel forward and backward speed (head dim 128, BF16) on A100-SXM 80G GPU. FlexAttention using PyTorch 2.6.0.dev20240920+cu124.

from 12.1% to 60.7%. FLASHMASK achieves 37.8% to 62.3% of the theoretical maximum FLOPs/s on the A100 GPU.

# 6 LIMITATIONS AND FUTURE DIRECTIONS

While FLASHMASK significantly advances the efficiency of attention mechanisms for long sequences, it has limitations. The column-wise mask representation reduces memory complexity from $O(N^2)$ to $O(N)$, offering substantial memory savings for long sequence training and effectively capturing the most common mask patterns. However, it cannot represent arbitrary masks, particularly those with irregular masked regions within a single column. Extreme cases, such as completely random masks, pose challenges for both representation and efficient computation. Future research should focus on developing more sophisticated sparse representations that simultaneously maximize expressiveness and computational efficiency, particularly those amenable to tiling techniques for high-performance kernels. Extending FLASHMASK to leverage features of newer architectures, such as NVIDIA's Hopper, could further enhance performance. Additionally, while our current implementation is based on the PaddlePaddle Ma et al. (2019) framework, integrating FLASHMASK into other popular deep learning frameworks could broaden its impact and accessibility. These efforts aim to address current limitations while expanding FLASHMASK's applicability across a broader range of tasks, contributing to the ongoing evolution of efficient transformer models for long sequence processing.

# 7 CONCLUSION

In this paper, we introduced FLASHMASK, an innovative extension of the FlashAttention algorithm that introduces a column-wise sparse mask representation to efficiently handle a wide spectrum of attention mask patterns in Transformer models. Our approach reduces the memory complexity from $O(N^2)$ to $O(N)$, enabling the processing of significantly longer sequences, which is crucial for modern large language models. By integrating this representation into the FlashAttention algorithm and implementing optimized kernels, FLASHMASK leverages sparsity in the attention mask to skip computations on fully masked blocks without sacrificing computational accuracy. This strategic approach achieves notable computational speedups, with observed end-to-end enhancements ranging from 1.65x to 3.22x during fine-tuning and alignment training of large language models, compared to the existing FlashAttention dense method. Furthermore, FLASHMASK significantly decreases the memory overhead associated with attention mask storage, thereby extending support for even longer sequence modeling. Additionally, FLASHMASK outperforms the latest counterpart, FlexAttention, by 12.1% to 60.7% in kernel TFLOPs/s, achieving 37.8% to 62.3% of the theoretical maximum FLOPs/s on the A100 GPU. Our approach has been validated on downstream tasks of large language models, and we anticipate its widespread adoption in the industry.

ACKNOWLEDGMENTS

This work was supported by the Beijing Municipal Science and Technology Project (No. Z231100010323002).

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

## A  APPENDIX

### A.1  BACKWARD PASS ALGORITHM DETAILS

The detailed implementation of the FLASHMASK backward pass is presented in Algorithm 2. Similar to the forward pass, we precompute the maximum and minimum values of **LTS**, **LTE**, **UTS**, and **UTE**. These precomputed values $LTStart_j^{min}$, $LTStart_j^{max}$, $LTEnd_j^{min}$, $LTEnd_j^{max}$, $UTStart_j^{min}$, $UTStart_j^{max}$, $UTEnd_j^{min}$, $UTEnd_j^{max}$ can be directly loaded into registers and kept resident because the backward computation operates in a column-parallel mode. Additionally, **LTS**$_j$, **LTE**$_j$, **UTS**$_j$, and **UTE**$_j$ can be loaded into SRAM outside of the inner loop (lines 10–11), thereby reducing the number of accesses to HBM to $4 \times N$. Within the inner loop, the computation logic of FLASHMASK remains identical to that of the forward pass.

### A.2  END-TO-END TRAINING THROUGHPUT

Recent models such as Llama 3.1 and GPT-4, the Claude series, and Google's Gemini support sequence modeling beyond 128K tokens. FLASHMASK, with its reduced memory overhead, facilitates training with even longer contexts. However, existing public DPO and RM datasets lack training data for scenarios exceeding 128K tokens. To comprehensively evaluate FLASHMASK, we constructed synthetic data to simulate long-sequence training and verify end-to-end throughput improvements. We validated our method across four downstream tasks involving the fine-tuning and alignment training of large language models: SFT, LoRA, DPO, and RM.

#### A.2.1  DATA CONSTRUCTION METHOD

For end-to-end training, to realistically simulate real dataset distributions, we needed to distinguish between source tokens and target tokens within a document's sequence length. Additionally, the data construction method differed from that used in the kernel experiments. Given a maximum training sequence length $N$ and a document count range $n \in [1, 10]$, we first randomly sampled the number of documents, then sampled each document's sequence length such that the total sequence length equaled $N$. The last document was considered as padding. For the RM training, special constraints

---

**Algorithm 2** FlashAttention-2 Backward Pass Extended with FLASHMASK

---

**Require:** Matrices $\mathbf{Q}, \mathbf{K}, \mathbf{V}, \mathbf{O}, \mathbf{dO} \in \mathbb{R}^{N \times d}$ in HBM, vector $L \in \mathbb{R}^N$ in HBM, block sizes $B_c$, $B_r$, vectors $\mathbf{LTS}, \mathbf{LTE}, \mathbf{UTS}, \mathbf{UTE} \in \mathbb{R}^N$.

1: Divide $\mathbf{Q}$ into $T_r = \left\lceil \frac{N}{B_r} \right\rceil$ blocks $\mathbf{Q}_1, \ldots, \mathbf{Q}_{T_r}$ of size $B_r \times d$ each, and divide $\mathbf{K}, \mathbf{V}$ in to $T_c = \left\lceil \frac{N}{B_c} \right\rceil$ blocks $\mathbf{K}_1, \ldots, \mathbf{K}_{T_c}$ and $\mathbf{V}_1, \ldots, \mathbf{V}_{T_c}$, of size $B_c \times d$ each.

2: Divide $\mathbf{O}$ into $T_r$ blocks $\mathbf{O}_1, \ldots, \mathbf{O}_{T_r}$ of size $B_r \times d$ each, divide $\mathbf{dO}$ into $T_r$ blocks $\mathbf{dO}_1, \ldots, \mathbf{dO}_{T_r}$ of size $B_r \times d$ each, and divide $L$ into $T_r$ blocks $L_1, \ldots, L_{T_r}$ of size $B_r$ each.

3: Initialize $\mathbf{dQ} = (0)_{N \times d}$ in HBM and divide it into $T_r$ blocks $\mathbf{dQ}_1, \ldots, \mathbf{dQ}_{T_r}$ of size $B_r \times d$ each. Divide $\mathbf{dK}, \mathbf{dV} \in \mathbb{R}^{N \times d}$ in to $T_c$ blocks $\mathbf{dK}_1, \ldots, \mathbf{dK}_{T_c}$ and $\mathbf{dV}_1, \ldots, \mathbf{dV}_{T_c}$, of size $B_c \times d$ each.

4: Compute $D = \text{rowsum}(\mathbf{dO} \circ \mathbf{O}) \in \mathbb{R}^d$ (pointwise multiply), write $D$ to HBM and divide it into $T_r$ blocks $D_1, \ldots, D_{T_r}$ of size $B_r$ each.

5: Divide $\mathbf{LTS}, \mathbf{LTE}, \mathbf{UTS}, \mathbf{UTE}$ into $T_c$ blocks $\mathbf{LTS}_1, \ldots, \mathbf{LTS}_{T_c}$, $\mathbf{LTE}_1, \ldots, \mathbf{LTE}_{T_c}$, $\mathbf{UTS}_1, \ldots, \mathbf{UTS}_{T_c}$, $\mathbf{UTE}_1, \ldots, \mathbf{UTE}_{T_c}$, of size $B_c$ each.

6: Precompute the min and max row index $LTStart_j^{min} = min(\mathbf{LTS}_j)$, $LTStart_j^{max} = max(\mathbf{LTS}_j)$, $LTEnd_j^{min} = min(\mathbf{LTE}_j)$, $LTEnd_j^{max} = max(\mathbf{LTE}_j)$, $UTStart_j^{min} = min(\mathbf{UTS}_j)$, $UTStart_j^{max} = max(\mathbf{UTS}_j)$, $UTEnd_j^{min} = min(\mathbf{UTE}_j)$, $UTEnd_j^{max} = max(\mathbf{UTE}_j)$, $\forall j \in \{1, \ldots, T_c\}$, write to HBM.

7: **for** $1 \leq j \leq T_c$ **do**

8:      Load $\mathbf{K}_j, \mathbf{V}_j$ from HBM to on-chip SRAM.

9:      Initialize $\mathbf{dK}_j = (0)_{B_c \times d}, \mathbf{dV}_j = (0)_{B_c \times d}$ on SRAM.

10:      Load $\mathbf{LTS}_j$ and $\mathbf{LTE}_j$ from HBM to on-chip SRAM.

11:      Load $\mathbf{UTS}_j$ and $\mathbf{UTE}_j$ from HBM to on-chip SRAM.

12:      **for** $1 \leq i \leq T_r$ **do**

13:          **if** $(i-1) \times B_r \geq LTStart_j^{max}$ **and** $i \times B_r \leq LTEnd_j^{min}$ **then**

14:              Continue // `lower triangular skip calculation of masked block`

15:          **end if**

16:          **if** $(i-1) \times B_r \geq UTStart_j^{max}$ **and** $i \times B_r \leq UTEnd_j^{min}$ **then**

17:              Continue // `upper triangular skip calculation of masked block`

18:          **end if**

19:          Load $\mathbf{Q}_i, \mathbf{O}_i, \mathbf{dO}_i, \mathbf{dQ}_i, L_i, D_i$ from HBM to on-chip SRAM.

20:          On chip, compute $\mathbf{S}_i^{(j)} = \mathbf{Q}_i \mathbf{K}_j^T \in \mathbb{R}^{B_r \times B_c}$.

21:          **if** $i \times B_r > LTStart_j^{min}$ **and** $(i-1) \times B_r < LTEnd_j^{max}$ **then**

22:              On chip, apply mask: $\mathbf{S}_i^{(j)}[x][y] = -\infty, \forall x, y$, such that $\mathbf{LTS}_j[y] \leq (i-1) \times B_r + x < \mathbf{LTE}_j[y]$

23:          **end if**

24:          **if** $i \times B_r > UTStart_j^{min}$ **and** $(i-1) \times B_r < UTEnd_j^{max}$ **then**

25:              On chip, apply mask: $\mathbf{S}_i^{(j)}[x][y] = -\infty, \forall x, y$, such that $\mathbf{UTS}_j[y] \leq (i-1) \times B_r + x < \mathbf{UTE}_j[y]$

26:          **end if**

27:          On chip, compute $\mathbf{P}_i^{(j)} = \exp(\mathbf{S}_{ij} - L_i) \in \mathbb{R}^{B_r \times B_c}$.

28:          On chip, compute $\mathbf{dV}_j \leftarrow \mathbf{dV}_j + (\mathbf{P}_i^{(j)})^\top \mathbf{dO}_i \in \mathbb{R}^{B_c \times d}$.

29:          On chip, compute $\mathbf{dP}_i^{(j)} = \mathbf{dO}_i \mathbf{V}_j^\top \in \mathbb{R}^{B_r \times B_c}$.

30:          On chip, compute $\mathbf{dS}_i^{(j)} = \mathbf{P}_i^{(j)} \circ (\mathbf{dP}_i^{(j)} - D_i) \in \mathbb{R}^{B_r \times B_c}$.

31:          Load $\mathbf{dQ}_i$ from HBM to SRAM, then on chip, update $\mathbf{dQ}_i \leftarrow \mathbf{dQ}_i + \mathbf{dS}_i^{(j)} \mathbf{K}_j \in \mathbb{R}^{B_r \times d}$, and write back to HBM.

32:          On chip, compute $\mathbf{dK}_j \leftarrow \mathbf{dK}_j + \mathbf{dS}_i^{(j)\top} \mathbf{Q}_i \in \mathbb{R}^{B_c \times d}$.

33:      **end for**

34:      Write $\mathbf{dK}_j, \mathbf{dV}_j$ to HBM.

35: **end for**

36: Return $\mathbf{dQ}, \mathbf{dK}, \mathbf{dV}$.

---

were applied: $n \in [1, 3]$ for $N \in (0, 4096]$ and $n \in [1, 4]$ for $N \in (4096, 8192]$. During sampling, we set the minimum document length to 128 for SFT, LoRA, and DPO, and 512 for RM. Padding lengths did not exceed 128 for SFT, LoRA, DPO, and 512 for RM.

Assuming a document had a sequence length $L$, it was further divided into source tokens and target tokens. SFT and LoRA were represented as (*Question, Answer*) pairs, DPO as (*Question, Answer*$_1$, *Answer*$_2$) with two answers, and RM, having 2 to 6 answers, was standard-

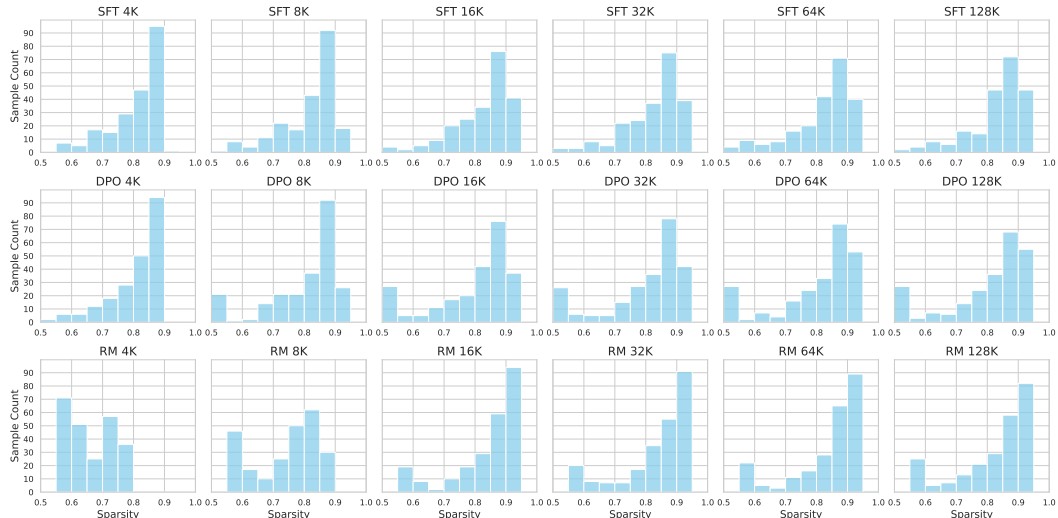

Figure 6: Sparsity distribution of synthetic dataset for end-to-end training throughput testing.

ized to have 6 answers: ($Question, Answer_1, \ldots, Answer_6$). Thus, $L$ was partitioned into a query and $k$ answers based on the training task, with $k$ being 1 for SFT and LoRA, 2 for DPO, and 6 for RM. The length of each answer was randomly determined from the range $\left[\frac{0.1L}{1+0.1k}, \frac{0.2L}{1+0.2k}\right]$, making each answer approximately 10% to 20% of the query length. Consequently, the query length was calculated as $L$ minus the total answer lengths. For each sequence length $N$, we collected 240 valid samples and categorized them into 10 bins by sparsity $\rho$, as illustrated in Figure 6.

Table 1: Hyperparameters and distributed configurations for various scales of Llama2 models.

| Model | LLama2-7B | LLama2-13B | LLama2-70B |
|---|---|---|---|
| Batch Size | 16 | 16 | 16 |
| AccSteps | 2 | 4 | 16 |
| Sharding Stage1 Degree | 8 | 4 | 1 |
| Tensor Parallel Degree | 4 | 4 | 8 |
| PipeLine Parallel Degree | 1 | 2 | 4 |
| Sequence Parallel | ✓ | ✓ | ✓ |

### A.2.2 EXPERIMENTAL CONFIGURATION AND DISTRIBUTED STRATEGY

We evaluated different model scales of Llama2 (7B, 13B, 70B) across sequence lengths ranging from 4K to 544K, comparing against two dense methods: Vanilla Attention and FlashAttention DenseMask. All end-to-end throughput experiments were conducted on four servers, each equipped with eight NVIDIA A800-SXM 80G GPUs, totaling 32 GPUs. The objective was not to optimize peak performance for each configuration but to assess scalability with longer sequences; thus, we uniformly enabled full recomputation. Model parameters and computations utilized the BFloat16 data type, while gradient accumulation and communication employed Float32. The hyperparameters and distributed strategies for different scales are detailed in Table 1.

### A.2.3 END-TO-END TRAINING MEMORY CONSUMPTION

Figure 2 in the main paper reports the end-to-end training throughput. We also recorded the peak memory consumption, presented in Figure 7. Notably, the memory usage of FLASHMASK increases significantly slower than that of dense methods. However, the figure also indicates that FLASHMASK's memory consumption still escalates rapidly with longer sequence lengths, primarily due to increased activation memory from longer sequences, as shown in Table 2. The *Param & Opt State* column indicates the memory consumption for parameters, gradients, and optimizer states, with sharding stage 1 applied. *Activations* refers to the memory consumed by the inputs of the 32 decoder layers. *Peak Mem One Layer* represents the peak memory usage when full recompute is employed. *Total* denotes the overall memory consumption for FlashAttention without the attention mask.

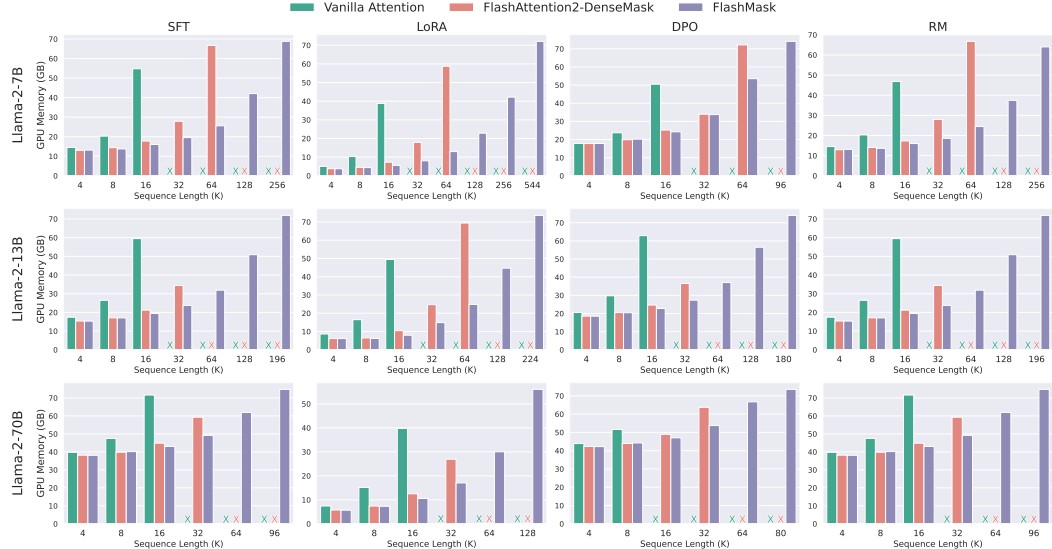

Figure 7: End-to-end training peak memory consumption across varying sequence lengths for different Llama2 model scales in four downstream training tasks: SFT, LoRA, DPO, and RM.

Table 2: Memory consumption comparison between FlashAttention without attention mask and FLASHMASK on the Llama-2 7B model. The observed differences in total memory footprint are attributed to memory fragmentation effects.

| Sequence Length (K) | Param & Opt State | Activations | Peak Mem One Layer | Total | FLASHMASK |
|---|---|---|---|---|---|
| 4 | 13.12 | 0.00 | 0.73 | 13.86 | 13.14 |
| 8 | 13.12 | 0.00 | 1.29 | 14.41 | 13.73 |
| 16 | 13.12 | 1.00 | 2.50 | 16.63 | 16.01 |
| 32 | 13.12 | 2.00 | 4.95 | 20.07 | 19.52 |
| 64 | 13.12 | 4.00 | 9.89 | 27.02 | 25.57 |
| 128 | 13.12 | 8.00 | 19.78 | 40.91 | 42.08 |
| 256 | 13.12 | 16.00 | 39.56 | 68.69 | 68.81 |

## A.3 END-TO-END TRAINING CONVERGENCE VERIFICATION

We selected the Llama 3.1 8B model to verify convergence across four downstream tasks involving the fine-tuning and alignment training of large language models: SFT, LoRA, DPO, and RM. SFT and LoRA utilized the same dataset, validated using `allenai/tulu-v2-sft-mixture` Ivison et al. (2023). For DPO and RM, which both employ (*Question*, *Answer*) data formats, we used the `HuggingFaceH4/ultrafeedback_binarized` Tunstall et al. (2023) dataset for validation. We consistently applied a linear learning rate decay strategy, with warm-up steps set to 3% of the total training steps. The AdamW optimizer was used with $\beta_1 = 0.9$ and $\beta_2 = 0.999$. Model parameters and computations utilized the BFloat16 data type, while gradient accumulation and communication employed Float32. The maximum training sequence length was set to 8K. Distributed parallelism combined sharding and tensor parallelism. Additional hyperparameters are listed in Table 3.

## A.4 SPARSITY-RELATED EXPERIMENTS

Prior to each run, we perform 10 warm-up iterations, followed by 100 runs of the kernel computation, recording the average execution time (in milliseconds) using CUDA Events. We aimed to verify that the computational complexity of FLASHMASK scales linearly with block sparsity in the attention mask. We report the total latency for the kernel's forward and backward passes. This validation was performed on sequences of length 32K for three common mask types: *Causal Document Mask*, *Share Question Mask*, and *Document Mask*, corresponding to downstream training tasks in large language models such as SFT, DPO/RM, and pre-training of vision models like NaViT, respectively.

Table 3: The configuration of end-to-end training loss convergence verification.

| Task | Dataset | Learning Rate | Training Step | Epochs | Batch Size | AccSteps | GPUs | Sharding Degree | TP Degree |
|------|---------|---------------|---------------|--------|------------|----------|------|-----------------|-----------|
| SFT | tulu-v2-sft-mixture | 2e-05 | 12000 | 3 | 16 | 1 | 32 | 16 | 2 |
| LoRA | tulu-v2-sft-mixture | 0.0002 | 12000 | 3 | 16 | 1 | 32 | 16 | 2 |
| DPO | HuggingFaceH4/ultrafeedback_binarized | 5e-07 | 4000 | 2 | 4 | 2 | 8 | 2 | 4 |
| RM | HuggingFaceH4/ultrafeedback_binarized | 1e-05 | 4000 | 2 | 4 | 1 | 8 | 4 | 2 |

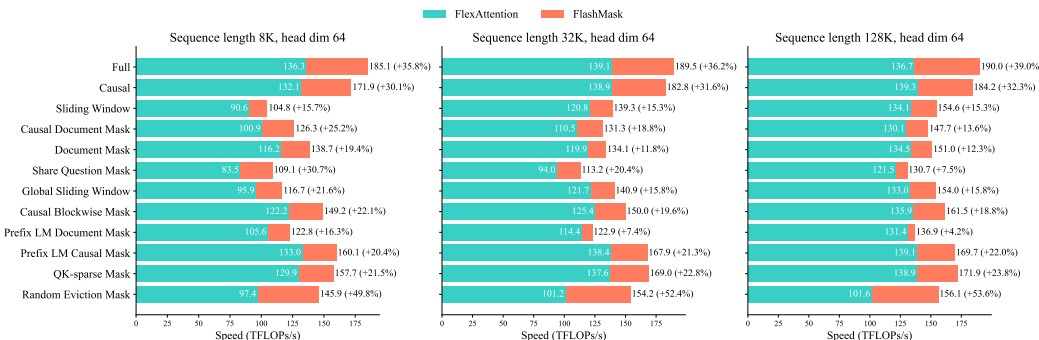

Figure 8: Kernel forward and backward speed (head dim 64, BF16) on A100-SXM 80G GPU. FlexAttention using PyTorch 2.6.0.dev20240920+cu124.

### A.4.1 DATA CONSTRUCTION METHOD

*Causal Document Mask* and *Share Question Mask* are causal attention types with block sparsity values in the range [0.5, 1.0], while the *Document Mask* is a bidirectional attention type with block sparsity values in [0.0, 1.0]. We partitioned the sparsity values into buckets: 10 buckets for causal types and 20 buckets for bidirectional types, each with intervals of 0.05, ensuring the number of samples per bucket ranged between 10 and 20.

For the *Causal Document Mask*, given a maximum sequence length, we limited the number of documents to [2, 20]. We randomly sampled the number of documents and then sampled the length of each document such that the total sequence length equaled the maximum sequence length. Following each sample, we calculated the block sparsity and assigned it to the corresponding bucket until each bucket met the required number of samples. The data sampling process for the *Document Mask* was similar; to ensure coverage of all sparsity levels, the number of documents was limited to [2, 10].

The data sampling for the *Share Question Mask* differed slightly. The number of documents was limited to [1, 5]. We first sampled the length of each document to sum up to the given maximum sequence length. Each document was then partitioned into a *Question* and *Answers*, ensuring there was one *Question* and 2 to 6 *Answers*. As before, after each sample, we calculated the block sparsity and allocated it to the appropriate bucket until all buckets were adequately populated.

In total, we sampled 182 samples for the *Causal Document Mask*, 175 for the *Share Question Mask*, and 374 for the *Document Mask*.

## A.5 KERNEL PERFORMANCE COMPARISON

### A.5.1 TESTING METHOD

Both FLASHMASK and FlexAttention exploit sparsity in the attention mask to skip fully masked blocks, thereby reducing redundant computations. To provide an intuitive comparison, we employ the TFLOPs/s metric for evaluation. For each test case, we assess both the forward and backward computations. Prior to each run, we perform 10 warm-up iterations, followed by 100 runs of the kernel computation, recording the average execution time (in milliseconds) using CUDA Events. Based on the block sparsity in the attention mask, we calculate the FLOPs for a single run and subsequently compute the TFLOPs/s.

### A.5.2 DATA CONSTRUCTION METHOD

We conducted detailed comparisons across varying batch sizes and sequence lengths (8K, 32K, 128K), different head dimensions (64, 128), and numbers of heads. We fixed the total number of tokens at

Table 4: Kernel speed details (8K, head dim 128, BF16) on A100-SXM 80G GPU.

| Method | Operation | FW Time (ms) | BW Time (ms) | TOTAL Time (ms) | FW TFLOPs | BW TFLOPs | TOTAL TFLOPs | FW TFLOPs/s | BW TFLOPs/s | TOTAL TFLOPs/s | Sparsity |
|---|---|---|---|---|---|---|---|---|---|---|---|
| FlexAttention | Full | 109.77 | 331.80 | 441.57 | 17.59 | 43.98 | 61.57 | 160.27 | 132.55 | 139.44 | 0.00 |
| | Causal | 55.61 | 179.82 | 235.43 | 8.93 | 22.33 | 31.27 | 160.65 | 124.20 | 132.81 | 0.49 |
| | Sliding Window | 10.20 | 41.85 | 52.05 | 1.33 | 3.33 | 4.66 | 130.50 | 79.54 | 89.53 | 0.92 |
| | Causal Document Mask | 17.57 | 62.99 | 80.56 | 2.42 | 6.05 | 8.47 | 136.79 | 95.10 | 104.16 | 0.86 |
| | Document Mask | 31.01 | 103.90 | 134.90 | 4.54 | 11.34 | 15.87 | 143.88 | 106.67 | 115.17 | 0.74 |
| | Share Question Mask | 13.17 | 50.07 | 63.24 | 1.62 | 4.05 | 5.68 | 122.39 | 80.38 | 89.11 | 0.91 |
| | Global Sliding Window | 21.65 | 79.82 | 101.47 | 2.89 | 7.24 | 10.13 | 133.74 | 90.67 | 99.85 | 0.84 |
| | Causal Blockwise Mask | 38.44 | 125.16 | 163.60 | 5.75 | 14.37 | 20.12 | 149.19 | 114.42 | 122.58 | 0.67 |
| | Prefix LM Document Mask | 18.92 | 66.00 | 84.92 | 2.53 | 6.32 | 8.85 | 132.59 | 94.89 | 103.27 | 0.86 |
| | Prefix LM Causal Mask | 69.06 | 218.47 | 287.53 | 11.06 | 27.66 | 38.72 | 160.20 | 126.61 | 134.68 | 0.37 |
| | QK-sparse Mask | 53.17 | 170.11 | 223.28 | 8.36 | 20.91 | 29.28 | 157.30 | 122.92 | 131.10 | 0.52 |
| | Random Eviction Mask | 67.53 | 215.56 | 283.09 | 8.93 | 22.33 | 31.27 | 132.30 | 103.61 | 110.45 | 0.49 |
| FLASHMASK | Full | 76.30 | 224.34 | 300.64 | 17.59 | 43.98 | 61.57 | 230.56 | 196.05 | 204.81 | 0.00 |
| | Causal | 39.01 | 118.60 | 157.61 | 8.93 | 22.33 | 31.27 | 229.02 | 188.31 | 198.39 | 0.49 |
| | Sliding Window | 8.83 | 30.59 | 39.41 | 1.33 | 3.33 | 4.66 | 150.84 | 108.83 | 118.24 | 0.92 |
| | Causal Document Mask | 13.78 | 44.24 | 58.03 | 2.42 | 6.05 | 8.47 | 174.26 | 135.48 | 144.67 | 0.86 |
| | Document Mask | 25.12 | 73.76 | 98.88 | 4.56 | 11.40 | 15.96 | 177.95 | 151.74 | 158.40 | 0.74 |
| | Share Question Mask | 9.83 | 33.87 | 43.70 | 1.62 | 4.05 | 5.68 | 163.93 | 118.89 | 129.01 | 0.91 |
| | Global Sliding Window | 20.10 | 53.07 | 73.17 | 2.89 | 7.24 | 10.13 | 144.05 | 136.36 | 138.47 | 0.84 |
| | Causal Blockwise Mask | 30.76 | 86.04 | 116.80 | 5.75 | 14.37 | 20.12 | 186.37 | 166.59 | 171.79 | 0.67 |
| | Prefix LM Document Mask | 16.04 | 46.82 | 62.86 | 2.53 | 6.32 | 8.85 | 156.17 | 133.89 | 139.58 | 0.86 |
| | Prefix LM Causal Mask | 55.20 | 162.31 | 217.51 | 11.06 | 27.66 | 38.72 | 200.42 | 170.42 | 178.03 | 0.37 |
| | QK-sparse Mask | 43.57 | 120.84 | 164.41 | 8.44 | 21.11 | 29.55 | 193.80 | 174.67 | 179.74 | 0.52 |
| | Random Eviction Mask | 49.04 | 135.06 | 184.10 | 8.93 | 22.33 | 31.27 | 182.17 | 165.36 | 169.84 | 0.49 |

Table 5: Kernel speed details (32K, head dim 128, BF16) on A100-SXM 80G GPU.

| Method | Operation | FW Time (ms) | BW Time (ms) | TOTAL Time (ms) | FW TFLOPs | BW TFLOPs | TOTAL TFLOPs | FW TFLOPs/s | BW TFLOPs/s | TOTAL TFLOPs/s | Sparsity |
|---|---|---|---|---|---|---|---|---|---|---|---|
| FlexAttention | Full | 434.91 | 1296.22 | 1731.12 | 70.37 | 175.92 | 246.29 | 161.80 | 135.72 | 142.27 | 0.00 |
| | Causal | 214.09 | 667.11 | 881.20 | 35.32 | 88.30 | 123.63 | 164.98 | 132.37 | 140.29 | 0.50 |
| | Sliding Window | 29.40 | 101.77 | 131.18 | 4.53 | 11.32 | 15.84 | 153.95 | 111.20 | 120.79 | 0.94 |
| | Causal Document Mask | 25.40 | 87.30 | 112.70 | 3.68 | 9.19 | 12.86 | 144.54 | 105.08 | 113.97 | 0.95 |
| | Document Mask | 39.29 | 126.59 | 165.88 | 5.72 | 14.29 | 20.01 | 145.78 | 111.30 | 118.98 | 0.92 |
| | Share Question Mask | 17.49 | 63.59 | 81.08 | 2.30 | 5.75 | 8.04 | 131.21 | 90.15 | 99.00 | 0.97 |
| | Global Sliding Window | 61.53 | 198.82 | 260.36 | 9.29 | 23.23 | 32.52 | 151.01 | 116.84 | 124.92 | 0.87 |
| | Causal Blockwise Mask | 59.26 | 187.03 | 246.29 | 8.94 | 22.35 | 31.29 | 150.30 | 118.95 | 126.48 | 0.87 |
| | Prefix LM Document Mask | 27.78 | 91.34 | 119.11 | 3.83 | 9.58 | 13.41 | 137.70 | 104.65 | 112.36 | 0.95 |
| | Prefix LM Causal Mask | 269.34 | 828.90 | 1098.24 | 44.05 | 110.12 | 154.17 | 163.55 | 132.85 | 140.38 | 0.37 |
| | QK-sparse Mask | 209.04 | 644.08 | 853.12 | 33.99 | 84.97 | 118.95 | 162.58 | 131.92 | 139.43 | 0.52 |
| | Random Eviction Mask | 261.30 | 810.44 | 1071.74 | 35.32 | 88.30 | 123.63 | 135.18 | 108.96 | 115.35 | 0.50 |
| FLASHMASK | Full | 304.25 | 860.73 | 1164.98 | 70.37 | 175.92 | 246.29 | 231.28 | 204.39 | 211.41 | 0.00 |
| | Causal | 153.25 | 430.64 | 583.89 | 35.32 | 88.30 | 123.63 | 230.49 | 205.05 | 211.73 | 0.50 |
| | Sliding Window | 28.50 | 72.25 | 100.76 | 4.53 | 11.32 | 15.84 | 158.83 | 156.63 | 157.25 | 0.94 |
| | Causal Document Mask | 24.76 | 60.51 | 85.27 | 3.68 | 9.19 | 12.86 | 148.12 | 151.60 | 150.59 | 0.95 |
| | Document Mask | 41.76 | 89.84 | 131.60 | 5.77 | 14.42 | 20.19 | 134.92 | 158.68 | 150.84 | 0.92 |
| | Share Question Mask | 18.28 | 42.77 | 61.05 | 2.30 | 5.75 | 8.04 | 125.41 | 134.07 | 131.47 | 0.97 |
| | Global Sliding Window | 65.36 | 140.86 | 206.21 | 9.29 | 23.23 | 32.52 | 142.17 | 164.92 | 157.71 | 0.87 |
| | Causal Blockwise Mask | 54.29 | 127.06 | 181.34 | 8.94 | 22.35 | 31.29 | 163.41 | 175.15 | 171.61 | 0.87 |
| | Prefix LM Document Mask | 33.37 | 64.23 | 97.59 | 3.83 | 9.58 | 13.41 | 114.50 | 148.85 | 137.07 | 0.95 |
| | Prefix LM Causal Mask | 217.99 | 606.89 | 824.88 | 44.05 | 110.12 | 154.17 | 202.07 | 181.45 | 186.90 | 0.37 |
| | QK-sparse Mask | 173.04 | 446.80 | 619.84 | 34.09 | 85.23 | 119.33 | 197.02 | 190.77 | 192.51 | 0.52 |
| | Random Eviction Mask | 192.18 | 494.40 | 686.59 | 35.32 | 88.30 | 123.63 | 183.79 | 178.61 | 180.06 | 0.50 |

128K; by varying the sequence length, we computed the corresponding batch size. With the hidden size fixed at 4096, varying the head dimension allowed us to determine the number of heads.

To encompass a broader range of block sparsity cases in the attention mask for a given sequence length, we utilized constructed data for testing. Given a test sequence length, we defined the document count range as $n \in [\text{Doc}_{min}, \text{Doc}_{max}]$. We first sampled the number of documents and then sampled the length of each document such that the total length equaled the test sequence length. For the *Share Question Mask* type, we further partitioned each document into one *Question* and 2 to 6 *Answers*. The document count ranges were [3, 7] for 8K, [10, 14] for 32K, and [11, 15] for 128K. For each sequence length, we generated five test data samples.

Table 6: Kernel speed details (128K, head dim 128, BF16) on A100-SXM 80G GPU.

| Method | Operation | FW Time (ms) | BW Time (ms) | TOTAL Time (ms) | FW TFLOPs | BW TFLOPs | TOTAL TFLOPs | FW TFLOPs/s | BW TFLOPs/s | TOTAL TFLOPs/s | Sparsity |
|---|---|---|---|---|---|---|---|---|---|---|---|
| FlexAttention | Full | 1726.14 | 5241.59 | 6967.73 | 281.48 | 703.69 | 985.16 | 163.07 | 134.25 | 141.39 | 0.00 |
| | Causal | 853.26 | 2647.03 | 3500.29 | 140.88 | 352.19 | 493.06 | 165.10 | 133.05 | 140.86 | 0.50 |
| | Sliding Window | 106.38 | 340.59 | 446.97 | 17.31 | 43.27 | 60.58 | 162.71 | 127.05 | 135.54 | 0.94 |
| | Causal Document Mask | 80.71 | 258.22 | 338.93 | 12.82 | 32.05 | 44.87 | 158.78 | 124.03 | 132.30 | 0.95 |
| | Document Mask | 159.54 | 494.81 | 654.35 | 25.68 | 64.21 | 89.89 | 160.44 | 129.20 | 136.81 | 0.91 |
| | Share Question Mask | 48.41 | 159.07 | 207.49 | 7.41 | 18.52 | 25.92 | 152.91 | 116.31 | 124.85 | 0.97 |
| | Global Sliding Window | 216.10 | 664.58 | 880.69 | 34.86 | 87.14 | 122.00 | 161.30 | 131.13 | 138.53 | 0.88 |
| | Causal Blockwise Mask | 221.27 | 671.55 | 892.82 | 35.32 | 88.29 | 123.60 | 159.60 | 131.46 | 138.44 | 0.87 |
| | Prefix LM Document Mask | 85.82 | 267.09 | 352.91 | 13.39 | 33.48 | 46.87 | 155.98 | 125.26 | 132.73 | 0.95 |
| | Prefix LM Causal Mask | 1073.03 | 3309.05 | 4382.08 | 175.99 | 439.98 | 615.97 | 164.01 | 132.96 | 140.56 | 0.37 |
| | QK-sparse Mask | 831.00 | 2556.67 | 3387.67 | 136.06 | 340.40 | 476.22 | 163.73 | 133.05 | 140.57 | 0.52 |
| | Random Eviction Mask | 1037.20 | 3318.50 | 4355.70 | 140.88 | 352.19 | 493.06 | 135.82 | 106.13 | 113.20 | 0.50 |
| FLASHMASK | Full | 1216.27 | 3403.06 | 4619.33 | 281.48 | 703.69 | 985.16 | 231.42 | 206.78 | 213.27 | 0.00 |
| | Causal | 631.12 | 1679.32 | 2310.44 | 140.88 | 352.19 | 493.06 | 223.21 | 209.72 | 213.41 | 0.50 |
| | Sliding Window | 105.84 | 238.96 | 344.74 | 17.31 | 43.27 | 60.58 | 163.63 | 181.09 | 175.73 | 0.94 |
| | Causal Document Mask | 87.46 | 179.89 | 267.35 | 12.82 | 32.05 | 44.87 | 146.27 | 178.03 | 167.61 | 0.95 |
| | Document Mask | 177.02 | 358.45 | 535.47 | 25.72 | 64.29 | 90.01 | 142.00 | 178.59 | 165.71 | 0.91 |
| | Share Question Mask | 62.65 | 109.80 | 172.44 | 7.41 | 18.52 | 25.92 | 118.01 | 168.51 | 150.12 | 0.97 |
| | Global Sliding Window | 248.39 | 482.82 | 731.21 | 34.86 | 87.14 | 122.00 | 140.33 | 180.49 | 166.85 | 0.88 |
| | Causal Blockwise Mask | 210.42 | 464.94 | 675.36 | 35.32 | 88.29 | 123.60 | 167.81 | 189.88 | 183.00 | 0.87 |
| | Prefix LM Document Mask | 121.43 | 193.09 | 314.52 | 13.39 | 33.48 | 46.87 | 110.01 | 173.27 | 148.75 | 0.95 |
| | Prefix LM Causal Mask | 891.90 | 2381.23 | 3273.13 | 175.99 | 439.98 | 615.97 | 197.32 | 184.77 | 188.19 | 0.37 |
| | QK-sparse Mask | 702.86 | 1748.06 | 2450.92 | 136.06 | 340.40 | 476.56 | 193.73 | 194.73 | 194.44 | 0.52 |
| | Random Eviction Mask | 776.92 | 1933.23 | 2710.16 | 140.88 | 352.19 | 493.06 | 181.32 | 182.18 | 181.93 | 0.50 |

### A.5.3 EXPERIMENTAL RESULTS

Figure 8 presents the comparison of total TFLOPs/s for forward and backward passes between FLASHMASK and FlexAttention when the head dimension is 64. Similar results are illustrated in Figure 5 in the main paper for a head dimension of 128. Across all cases, FLASHMASK outperforms FlexAttention in terms of total TFLOPs/s for both forward and backward passes, with improvements

Table 7: Kernel speed details (8K, head dim 64, BF16) on A100-SXM 80G GPU.

| Method | Operation | FW Time (ms) | BW Time (ms) | TOTAL Time (ms) | FW TFLOPs | BW TFLOPs | TOTAL TFLOPs | FW TFLOPs/s | BW TFLOPs/s | TOTAL TFLOPs/s | Sparsity |
|---|---|---|---|---|---|---|---|---|---|---|---|
| FlexAttention | Full | 111.99 | 339.81 | 451.80 | 17.59 | 43.98 | 61.57 | 157.09 | 129.43 | 136.28 | 0.00 |
| | Causal | 56.41 | 180.30 | 236.71 | 8.93 | 22.33 | 31.27 | 158.37 | 123.87 | 132.09 | 0.49 |
| | Sliding Window | 10.69 | 40.74 | 51.43 | 1.33 | 3.33 | 4.66 | 124.51 | 81.71 | 90.61 | 0.92 |
| | Causal Document Mask | 18.03 | 65.14 | 83.17 | 2.42 | 6.05 | 8.47 | 133.31 | 91.92 | 100.85 | 0.86 |
| | Document Mask | 31.17 | 102.82 | 133.98 | 4.54 | 11.34 | 15.87 | 143.68 | 107.97 | 116.20 | 0.74 |
| | Share Question Mask | 13.79 | 53.84 | 67.63 | 1.62 | 4.06 | 5.68 | 117.11 | 74.87 | 83.46 | 0.91 |
| | Global Sliding Window | 23.83 | 81.80 | 105.63 | 2.89 | 7.24 | 10.13 | 121.49 | 88.47 | 95.92 | 0.84 |
| | Causal Blockwise Mask | 39.45 | 124.74 | 164.18 | 5.75 | 14.37 | 20.12 | 145.32 | 114.83 | 122.15 | 0.67 |
| | Prefix LM Document Mask | 18.49 | 64.64 | 83.13 | 2.53 | 6.32 | 8.85 | 135.95 | 96.92 | 105.57 | 0.86 |
| | Prefix LM Causal Mask | 70.71 | 220.50 | 291.21 | 11.06 | 27.66 | 38.72 | 156.47 | 125.44 | 132.98 | 0.37 |
| | QK-sparse Mask | 54.11 | 171.32 | 225.43 | 8.36 | 20.91 | 29.28 | 154.57 | 122.05 | 129.86 | 0.52 |
| | Random Eviction Mask | 75.03 | 246.11 | 321.14 | 8.93 | 22.33 | 31.27 | 119.07 | 90.75 | 97.36 | 0.49 |
| FLASHMASK | Full | 87.04 | 245.56 | 332.59 | 17.59 | 43.98 | 61.57 | 202.13 | 179.10 | 185.13 | 0.00 |
| | Causal | 45.31 | 136.58 | 181.89 | 8.93 | 22.33 | 31.27 | 197.18 | 163.52 | 171.90 | 0.49 |
| | Sliding Window | 10.39 | 34.06 | 44.45 | 1.33 | 3.33 | 4.66 | 128.14 | 97.73 | 104.84 | 0.92 |
| | Causal Document Mask | 16.08 | 50.34 | 66.42 | 2.42 | 6.05 | 8.47 | 149.08 | 119.03 | 126.31 | 0.86 |
| | Document Mask | 28.11 | 84.48 | 112.59 | 4.56 | 11.40 | 15.96 | 158.54 | 132.11 | 138.71 | 0.74 |
| | Share Question Mask | 12.69 | 39.06 | 51.75 | 1.62 | 4.06 | 5.68 | 127.11 | 103.22 | 109.07 | 0.91 |
| | Global Sliding Window | 23.68 | 63.17 | 86.85 | 2.89 | 7.24 | 10.13 | 122.25 | 114.57 | 116.66 | 0.84 |
| | Causal Blockwise Mask | 34.22 | 100.23 | 134.44 | 5.75 | 14.37 | 20.12 | 167.38 | 142.95 | 149.17 | 0.67 |
| | Prefix LM Document Mask | 17.99 | 53.44 | 71.42 | 2.53 | 6.32 | 8.85 | 139.18 | 117.24 | 122.77 | 0.86 |
| | Prefix LM Causal Mask | 59.65 | 182.22 | 241.87 | 11.06 | 27.66 | 38.72 | 185.47 | 151.79 | 160.10 | 0.37 |
| | QK-sparse Mask | 47.46 | 139.89 | 187.35 | 8.44 | 21.11 | 29.55 | 177.89 | 150.89 | 157.73 | 0.52 |
| | Random Eviction Mask | 57.58 | 156.77 | 214.35 | 8.93 | 22.33 | 31.27 | 155.15 | 142.47 | 145.87 | 0.49 |

ranging from 4.2% to 53.6%. FLASHMASK achieves 33.6% to 55.1% of the theoretical maximum FLOPs/s on the A100 GPU. Tables 4 – 9 detail, for each test mask case, the sparsity, and the forward and backward computation latency, TFLOPs, and TFLOPs/s.

Table 8: Kernel speed details (32K, head dim 64, BF16) on A100-SXM 80G GPU.

| Method | Operation | FW Time (ms) | BW Time (ms) | TOTAL Time (ms) | FW TFLOPs | BW TFLOPs | TOTAL TFLOPs | FW TFLOPs/s | BW TFLOPs/s | TOTAL TFLOPs/s | Sparsity |
|---|---|---|---|---|---|---|---|---|---|---|---|
| FlexAttention | Full | 445.55 | 1325.12 | 1770.67 | 70.37 | 175.92 | 246.29 | 157.94 | 132.76 | 139.09 | 0.00 |
| | Causal | 218.23 | 671.98 | 890.21 | 35.32 | 88.30 | 123.63 | 161.85 | 131.41 | 138.87 | 0.50 |
| | Sliding Window | 30.15 | 100.98 | 131.13 | 4.53 | 11.32 | 15.84 | 150.13 | 112.08 | 120.83 | 0.94 |
| | Causal Document Mask | 26.17 | 90.05 | 116.22 | 3.68 | 9.19 | 12.86 | 140.30 | 101.85 | 110.50 | 0.95 |
| | Document Mask | 38.89 | 125.98 | 164.86 | 5.72 | 14.29 | 20.01 | 145.94 | 111.98 | 119.94 | 0.92 |
| | Share Question Mask | 18.75 | 69.39 | 88.14 | 2.37 | 5.93 | 8.31 | 126.37 | 85.32 | 94.04 | 0.97 |
| | Global Sliding Window | 64.43 | 202.92 | 267.34 | 9.29 | 23.23 | 32.52 | 144.23 | 114.48 | 121.65 | 0.87 |
| | Causal Blockwise Mask | 61.11 | 187.21 | 248.32 | 8.94 | 22.35 | 31.29 | 145.71 | 118.84 | 125.45 | 0.87 |
| | Prefix LM Document Mask | 26.91 | 90.11 | 117.02 | 3.83 | 9.58 | 13.41 | 142.18 | 106.09 | 114.39 | 0.95 |
| | Prefix LM Causal Mask | 278.20 | 836.05 | 1114.25 | 44.05 | 110.12 | 154.17 | 158.33 | 131.72 | 138.36 | 0.37 |
| | QK-sparse Mask | 215.41 | 648.92 | 864.33 | 33.99 | 84.97 | 118.95 | 157.77 | 130.93 | 137.62 | 0.52 |
| | Random Eviction Mask | 292.99 | 928.88 | 1221.87 | 35.32 | 88.30 | 123.63 | 120.56 | 95.07 | 101.18 | 0.50 |
| FLASHMASK | Full | 346.27 | 953.51 | 1299.78 | 70.37 | 175.92 | 246.29 | 203.22 | 184.50 | 189.49 | 0.00 |
| | Causal | 175.61 | 500.70 | 676.31 | 35.32 | 88.30 | 123.63 | 201.13 | 176.36 | 182.79 | 0.50 |
| | Sliding Window | 32.34 | 81.43 | 113.77 | 4.53 | 11.32 | 15.84 | 139.97 | 138.98 | 139.27 | 0.94 |
| | Causal Document Mask | 28.69 | 69.11 | 97.80 | 3.68 | 9.19 | 12.86 | 127.83 | 132.73 | 131.29 | 0.95 |
| | Document Mask | 44.79 | 103.14 | 147.93 | 5.77 | 14.42 | 20.19 | 127.83 | 138.00 | 134.12 | 0.92 |
| | Share Question Mask | 22.92 | 50.30 | 73.22 | 2.37 | 5.93 | 8.31 | 103.30 | 117.71 | 113.19 | 0.97 |
| | Global Sliding Window | 69.46 | 161.42 | 230.88 | 9.29 | 23.23 | 32.52 | 133.78 | 143.91 | 140.86 | 0.87 |
| | Causal Blockwise Mask | 59.67 | 147.83 | 207.50 | 8.94 | 22.35 | 31.29 | 148.59 | 150.60 | 149.98 | 0.87 |
| | Prefix LM Document Mask | 35.43 | 73.44 | 108.87 | 3.83 | 9.58 | 13.41 | 107.83 | 130.17 | 122.87 | 0.95 |
| | Prefix LM Causal Mask | 234.73 | 683.57 | 918.30 | 44.05 | 110.12 | 154.17 | 187.66 | 161.10 | 167.89 | 0.37 |
| | QK-sparse Mask | 185.20 | 520.75 | 705.95 | 34.09 | 85.23 | 119.33 | 184.09 | 163.67 | 169.03 | 0.52 |
| | Random Eviction Mask | 223.56 | 578.42 | 801.98 | 35.32 | 88.30 | 123.63 | 158.00 | 152.66 | 154.15 | 0.50 |

Table 9: Kernel speed details (128K, head dim 64, BF16) on A100-SXM 80G GPU.

| Method | Operation | FW Time (ms) | BW Time (ms) | TOTAL Time (ms) | FW TFLOPs | BW TFLOPs | TOTAL TFLOPs | FW TFLOPs/s | BW TFLOPs/s | TOTAL TFLOPs/s | Sparsity |
|---|---|---|---|---|---|---|---|---|---|---|---|
| FlexAttention | Full | 1779.78 | 5424.93 | 7204.72 | 281.48 | 703.69 | 985.16 | 158.15 | 129.71 | 136.74 | 0.00 |
| | Causal | 873.21 | 2666.96 | 3540.16 | 140.88 | 352.19 | 493.06 | 161.33 | 132.06 | 139.28 | 0.50 |
| | Sliding Window | 108.08 | 343.79 | 451.88 | 17.31 | 43.27 | 60.58 | 160.14 | 125.87 | 134.06 | 0.94 |
| | Causal Document Mask | 83.21 | 261.48 | 344.69 | 12.82 | 32.05 | 44.87 | 154.02 | 122.48 | 130.09 | 0.95 |
| | Document Mask | 164.96 | 501.19 | 666.15 | 25.68 | 64.21 | 89.89 | 155.36 | 127.63 | 134.49 | 0.91 |
| | Share Question Mask | 50.28 | 162.70 | 212.97 | 7.40 | 18.51 | 25.91 | 147.12 | 113.60 | 121.51 | 0.97 |
| | Global Sliding Window | 228.79 | 688.21 | 917.00 | 34.86 | 87.14 | 122.00 | 152.35 | 126.62 | 133.04 | 0.88 |
| | Causal Blockwise Mask | 226.86 | 682.44 | 909.31 | 35.32 | 88.29 | 123.60 | 155.66 | 129.37 | 135.93 | 0.87 |
| | Prefix LM Document Mask | 87.26 | 269.21 | 356.47 | 13.39 | 33.48 | 46.87 | 153.43 | 124.29 | 131.42 | 0.95 |
| | Prefix LM Causal Mask | 1093.12 | 3335.61 | 4428.74 | 175.99 | 439.98 | 615.97 | 161.00 | 131.90 | 139.08 | 0.37 |
| | QK-sparse Mask | 846.40 | 2583.14 | 3429.54 | 136.06 | 340.15 | 476.22 | 160.75 | 131.68 | 138.86 | 0.52 |
| | Random Eviction Mask | 1167.58 | 3683.61 | 4851.20 | 140.88 | 352.19 | 493.06 | 120.66 | 95.61 | 101.64 | 0.50 |
| FLASHMASK | Full | 1383.21 | 3800.81 | 5184.02 | 281.48 | 703.69 | 985.16 | 203.49 | 185.14 | 190.04 | 0.00 |
| | Causal | 706.68 | 1970.00 | 2676.68 | 140.88 | 352.19 | 493.06 | 199.35 | 178.78 | 184.21 | 0.50 |
| | Sliding Window | 120.53 | 271.26 | 391.79 | 17.31 | 43.27 | 60.58 | 143.60 | 159.52 | 154.63 | 0.94 |
| | Causal Document Mask | 98.31 | 204.97 | 303.28 | 12.82 | 32.05 | 44.87 | 130.11 | 156.25 | 147.75 | 0.95 |
| | Document Mask | 182.42 | 406.31 | 588.74 | 25.72 | 64.29 | 90.01 | 137.92 | 157.52 | 150.97 | 0.91 |
| | Share Question Mask | 72.72 | 125.14 | 197.87 | 7.40 | 18.51 | 25.91 | 101.51 | 147.72 | 130.66 | 0.97 |
| | Global Sliding Window | 243.70 | 548.38 | 792.08 | 34.86 | 87.14 | 122.00 | 143.03 | 158.91 | 154.03 | 0.88 |
| | Causal Blockwise Mask | 226.04 | 539.11 | 765.14 | 35.32 | 88.29 | 123.60 | 156.21 | 163.76 | 161.53 | 0.87 |
| | Prefix LM Document Mask | 123.12 | 218.68 | 341.80 | 13.39 | 33.48 | 46.87 | 108.50 | 153.00 | 136.90 | 0.95 |
| | Prefix LM Causal Mask | 943.12 | 2686.99 | 3630.11 | 175.99 | 439.98 | 615.97 | 186.60 | 163.74 | 169.68 | 0.37 |
| | QK-sparse Mask | 736.38 | 2036.58 | 2772.96 | 136.16 | 340.40 | 476.56 | 184.91 | 167.14 | 171.86 | 0.52 |
| | Random Eviction Mask | 893.22 | 2265.78 | 3159.00 | 140.88 | 352.19 | 493.06 | 157.72 | 155.44 | 156.08 | 0.50 |

# B FLASHMASK APPLICATION IN INFERENCE

In the main body of our paper, we focused on the application of FLASHMASK during the training phase of large-scale models. However, it is important to highlight that FLASHMASK is equally effective during the inference stage. In this appendix, we provide detailed experimental results demonstrating the efficacy of FLASHMASK in inference, comparing it with state-of-the-art attention implementations, including FlashInfer Ye et al. (2025).

## B.1 EXPERIMENTAL SETUP

Our experiments were conducted on an NVIDIA A100-SXM 80G GPU using FlashInfer version 0.1.6, CUDA 12.1, PyTorch 2.4, and BF16 data type. We set the batch size to 1, with 32 query/output

Table 10: Performance Comparison on **Causal Document Mask** at 8K, 32K, and 128K Sequence Lengths

| Method | Seq Length | Sparsity | FW Time (ms) | FW TFLOPs | FW TFLOPs/s |
|---|---|---|---|---|---|
| FlashInfer SparseMask | 8,192 | 0.8806 | 9.33 | 0.1313 | 13.95 |
| FlashInfer DenseMask | 8,192 | 0.8806 | 11.93 | 0.1313 | 11.01 |
| FLASHMASK | 8,192 | 0.8806 | 0.96 | 0.1313 | 135.07 |
| FlashInfer SparseMask | 32,768 | 0.9532 | 54.77 | 0.8233 | 15.01 |
| FlashInfer DenseMask | 32,768 | 0.9532 | 184.20 | 0.8233 | 4.47 |
| FLASHMASK | 32,768 | 0.9532 | 5.99 | 0.8233 | 137.09 |
| FlashInfer SparseMask | 131,072 | 0.9558 | 788.94 | 12.4435 | 15.77 |
| FlashInfer DenseMask | 131,072 | 0.9558 | 2,948.23 | 12.4435 | 4.22 |
| FLASHMASK | 131,072 | 0.9558 | 84.13 | 12.4435 | 147.58 |

Table 11: Performance Comparison on **Shared Question Mask** at 8K, 32K, and 128K Sequence Lengths

| Method | Seq Length | Sparsity | FW Time (ms) | FW TFLOPs | FW TFLOPs/s |
|---|---|---|---|---|---|
| FlashInfer SparseMask | 8,192 | 0.9324 | 6.12 | 0.0743 | 11.76 |
| FlashInfer DenseMask | 8,192 | 0.9324 | 11.94 | 0.0743 | 6.23 |
| FLASHMASK | 8,192 | 0.9324 | 0.73 | 0.0743 | 98.49 |
| FlashInfer SparseMask | 32,768 | 0.9742 | 32.87 | 0.4537 | 13.74 |
| FlashInfer DenseMask | 32,768 | 0.9742 | 184.40 | 0.4537 | 2.46 |
| FLASHMASK | 32,768 | 0.9742 | 4.59 | 0.4537 | 98.26 |
| FlashInfer SparseMask | 131,072 | 0.9751 | 443.80 | 7.0146 | 15.80 |
| FlashInfer DenseMask | 131,072 | 0.9751 | 2,948.89 | 7.0146 | 2.38 |
| FLASHMASK | 131,072 | 0.9751 | 61.57 | 7.0146 | 113.21 |

heads and 8 key/value heads, each with a head dimension of 128. The evaluation included typical attention masks such as the *Causal Document Mask*, *Document Mask*, and *Shared Question Mask*.

To ensure compatibility with FlashInfer's sparse mask representation (where the mask block size $C = 64$), we adapted the datasets from Section A.5.2 so that each sub-document sequence length is divisible by 64. We defined the *mask block size* based on FlashInfer's Block Sparse Row (BSR) API parameters $R$ and $C$, and matched the *tiling block size* to the kernel's operational dimensions.

### B.2 COMPARISON WITH FLASHINFER

We compared FLASHMASK with FlashInfer's dense mask API (`single_prefill_with_kv_cache`) and sparse mask API (`BlockSparseAttentionWrapper`) across various sequence lengths (8K, 32K, and 128K tokens). The results are summarized in Tables 10 to 14.

**Efficiency Analysis:** FLASHMASK consistently outperformed both the dense and sparse implementations of FlashInfer in terms of TFLOPs/s, particularly addressing the inefficiencies observed with FlashInfer's dense mask API. While FlashInfer with sparse masks showed performance gains with increasing mask block sizes ($R, C \geq 16$), such large block sizes are seldom practical due to the nature of attention patterns in real-world applications.

In the FlashInfer `single_prefill_with_kv_cache` implementation (see `prefill.cuh` lines 1234–1241[3]), token-by-token dense masks lead to significant inefficiencies by performing unnecessary computations on fully masked blocks. Additionally, in FlashInfer's `BlockSparseAttentionWrapper`, smaller mask block sizes increase the padded batch size (`nblks(padded_batch_size, 1, num_kv_heads)`), negatively impacting performance due to suboptimal kernel hyper-parameter tuning. In contrast, FLASHMASK efficiently computes only the required tiling blocks, avoiding redundant calculations, and thus achieves superior TFLOPs/s.

---

[3] `https://github.com/flashinfer-ai/flashinfer/blob/v0.1.6/include/flashinfer/attention/prefill.cuh#L1234-L1241`

## B.3 IMPACT OF MASK BLOCK SIZE

For the *Document Mask*, we investigated the effect of varying the mask block size ($R/C$) on performance. Although FlashInfer DenseMask and FLASHMASK do not utilize specific $R/C$ values, we included them in the comparison for completeness. The results for sequence lengths of 8K, 32K, and 128K tokens are presented in Tables 12, 13, and 14, respectively.

Table 12: Performance on **Document Mask** at 8K Sequence Length with Varying Mask Block Sizes

| Method | $R/C$ | Sparsity | FW Time (ms) | FW TFLOPs | FW TFLOPs/s |
|---|---|---|---|---|---|
| FlashInfer SparseMask | 1 | 0.7868 | 15.39 | 0.2344 | 15.19 |
| FlashInfer SparseMask | 2 | 0.7613 | 8.57 | 0.2624 | 30.48 |
| FlashInfer SparseMask | 4 | 0.7613 | 4.31 | 0.2624 | 60.57 |
| FlashInfer SparseMask | 8 | 0.7613 | 3.23 | 0.2624 | 80.97 |
| FlashInfer SparseMask | 16 | 0.7613 | 1.65 | 0.2624 | 158.55 |
| FlashInfer SparseMask | 32 | 0.7613 | 1.51 | 0.2624 | 172.61 |
| FlashInfer SparseMask | 64 | 0.7613 | 1.51 | 0.2624 | 172.82 |
| FlashInfer DenseMask | – | 0.7613 | 11.91 | 0.2624 | 22.03 |
| FLASHMASK | – | 0.7613 | 1.66 | 0.2624 | 156.82 |

Table 13: Performance on **Document Mask** at 32K Sequence Length with Varying Mask Block Sizes

| Method | $R/C$ | Sparsity | FW Time (ms) | FW TFLOPs | FW TFLOPs/s |
|---|---|---|---|---|---|
| FlashInfer SparseMask | 1 | 0.9064 | 104.65 | 1.6460 | 15.73 |
| FlashInfer SparseMask | 2 | 0.9064 | 52.46 | 1.6460 | 31.36 |
| FlashInfer SparseMask | 4 | 0.9064 | 25.96 | 1.6460 | 63.47 |
| FlashInfer SparseMask | 8 | 0.9064 | 19.68 | 1.6460 | 83.59 |
| FlashInfer SparseMask | 16 | 0.9064 | 9.87 | 1.6460 | 166.58 |
| FlashInfer SparseMask | 32 | 0.9064 | 8.89 | 1.6460 | 185.13 |
| FlashInfer SparseMask | 64 | 0.9064 | 8.89 | 1.6460 | 185.16 |
| FlashInfer DenseMask | – | 0.9064 | 183.99 | 1.6460 | 8.95 |
| FLASHMASK | – | 0.9064 | 11.73 | 1.6460 | 139.84 |

Table 14: Performance on **Document Mask** at 128K Sequence Length with Varying Mask Block Sizes

| Method | $R/C$ | Sparsity | FW Time (ms) | FW TFLOPs | FW TFLOPs/s |
|---|---|---|---|---|---|
| FlashInfer SparseMask | 1 | 0.9116 | 1,571.12 | 24.8848 | 15.84 |
| FlashInfer SparseMask | 2 | 0.9116 | 783.62 | 24.8848 | 31.75 |
| FlashInfer SparseMask | 4 | 0.9116 | 391.20 | 24.8848 | 63.61 |
| FlashInfer SparseMask | 8 | 0.9116 | 288.97 | 24.8848 | 86.11 |
| FlashInfer SparseMask | 16 | 0.9116 | 145.13 | 24.8848 | 171.45 |
| FlashInfer SparseMask | 32 | 0.9116 | 131.31 | 24.8848 | 189.50 |
| FlashInfer SparseMask | 64 | 0.9116 | 131.33 | 24.8848 | 189.48 |
| FlashInfer DenseMask | – | 0.9116 | 2,946.81 | 24.8848 | 8.44 |
| FLASHMASK | – | 0.9116 | 172.81 | 24.8848 | 143.68 |

