# OpenReview forum: "FlashMask: Efficient and Rich Mask Extension of FlashAttention"
_ICLR.cc/2025/Conference — ICLR 2025 Poster_

### Official Review · Reviewer_gKFL · 2024-10-25

**Soundness:** 3
**Presentation:** 3
**Contribution:** 3
**Rating:** 8
**Confidence:** 4

**Summary:**

The article presents Flash Mask - a method to incorporate a wide class of attention masks into flash attention. Algorithmically, this means that the  non trivial structuring of the mask is incorporated into the online calculation of the softmax operation involved in self attention without materializing the full mask and paying quadratic in sequence length memory. Furthermore, this algorithm is implemented in a hardware aware fashion, much like flash attention to minimize memory access and data movement while exploiting the thread group level parallelism in GPUs. Empirically, this method is benchmarked against the dense mask of flash attention 2 and also against flex attention - an alternative state of the art method to incorporate structured masks in efficient attention computations and the method presented shows noticeable gains both in inference and in training.

**Strengths:**

The strengths of this paper are in the novelty of the incorporation of a class of structured masks into the online softmax calculation involved in memory efficient attention and further into the hardware aware version of the algorithm which is flash attention, and in the  comprehensive validation of the superior efficiency of the algorithm when benchmarked against dense masking in conventional flash attention 2 and in flex attention.
The algorithm for the forward and backward pass are presented clearly and the empirical results are presented clearly.

**Weaknesses:**

As admitted by the authors, this method cannot handle irregular masking patterns within a column of the mask, or completely arbitrary masking patterns.

**Questions:**

Following up on the limitations, I would like the know whether the authors think that arbitrary masking patterns can be incorporated in a GPU friendly manner or if memory efficient implementations of such masking patterns would require alternative hardware architectures (such as those developed by Cerebras).

---

> ### Author Response · Authors · 2024-11-22
>
> Dear Reviewer,
>
> Thank you for your insightful evaluation and the positive feedback on our work. We are pleased to hear your interest in the potential extension of FlashMask's capabilities.
>
> **Weaknesses:** As admitted by the authors, this method cannot handle irregular masking patterns within a column of the mask, or completely arbitrary masking patterns.
>
> While it is true that FlashMask does not currently support arbitrary masking patterns, it effectively manages the common mask types utilized in LLM training. We are dedicated to enhancing the expressiveness of FlashMask by addressing some limitations inherent in the current two range-based ($LTS$, $LTE$, $UTS$, $UTE$) sparse representation. Specifically, we aim to improve its capability to handle multi range-based masks and introduce expression-based descriptions, which would remove the necessity for explicit attention mask inputs. For instance, if an attention mask can be described using a formula or expression, it potentially eliminates the need for explicit mask data by allowing the kernel to compute mask positions directly. Furthermore, we are focused on refining FlashMask's implementation to reduce unnecessary memory access and condition checks, particularly in relation to fully masked blocks, thus further enhancing its performance.
>
>
> **Q: Following up on the limitations, I would like to know whether the authors think that arbitrary masking patterns can be incorporated in a GPU-friendly manner or if memory-efficient implementations of such masking patterns would require alternative hardware architectures (such as those developed by Cerebras).**
>
> A: Based on the information we found on the [Cerebras](https://cerebras.ai) website, Cerebras' hardware architecture features large global memory, extensive SRAM, and high memory bandwidth. Theoretically, these characteristics allow it to incorporate arbitrary masking patterns more effectively than traditional GPUs.
>
> Thank you once again for your constructive feedback, which continues to guide our development efforts.
>
> Best regards

---

> > ### Comment · Reviewer_gKFL · 2024-11-26
> >
> > I appreciate the authors' clarifications and acknowledge that their method is more than adequate to handle the kinds of masking patterns commonly used in LLM training.

---

### Official Review · Reviewer_gr4R · 2024-10-29

**Soundness:** 3
**Presentation:** 3
**Contribution:** 3
**Rating:** 8
**Confidence:** 4

**Summary:**

This paper introduces an extension for SDPA that supports different types of masks in an easy-to-understand way. The method is novel due to its sparse representation. Experiments show FlashMask outperforms FlexAttention by a significant gap.

**Strengths:**

Although this representation might be similar to COO/CSR/CSC, this is the first time I have ever seen these techniques used in attention, one of the most important operators in LLMs.

**Weaknesses:**

This paper lacks two baselines:
1.  Flashinfer with dense masks;
2.  Flashinfer sparse mask (https://docs.flashinfer.ai/api/python/sparse.html);

Although Flashinfer does not support backward, I believe it is an important baseline for SOTA attention implementation. If this comparison is presented, I will raise my score.

**Questions:**

1. How can this technique be integrated with page attention?
2. Can tree-based speculative decoding benefit from this customized attention?
3. Can you report evaluation results on machines such as P100, V100, A10G, and H100? (other than A100)

**Details Of Ethics Concerns:**

No.

---

> ### Author Response · Authors · 2024-11-22
>
> Dear Reviewer,
>
> Thank you for engaging with our work and for highlighting the importance of extending evaluation to include FlashInfer as a baseline. We greatly appreciate your insights, which aid in advancing our research on FlashMask.
>
> **Response to Weaknesses and Questions:**
>
> We acknowledge the value that FlashInfer brings, bridging training and inference applications. Should our paper be accepted at ICLR 2025, we plan to reference FlashInfer in the camera-ready version and include the following experimental results in the appendix. Our primary focus with FlashMask has been addressing the extensive high bandwidth memory (HBM) requirements posed by diverse attention mask types in large-scale model training. Nevertheless, FlashMask's capabilities extend into inference stages as well, warranting consideration among leading attention mechanisms.
>
>
> We defined the "mask block size" based on FlashInfer's description, utilizing the BSR API's parameters R and C. The "tiling block size" was designed to match the operational dimensions of the kernel. Our experiments were conducted on an A100-SXM 80G GPU using the official version of FlashInfer v0.1.6, with CUDA 12.1, PyTorch 2.4, and BF16 as the data type. The configuration settings included `batch_size = 1`, `num_qo_heads = 32`, `num_kv_heads = 8`, and `head_dim = 128`. We compared FlashMask against the dense mask API `single_prefill_with_kv_cache` and the sparse mask API `BlockSparseAttentionWrapper` (with varying R/C values) in FlashInfer. Typical attention masks, such as the Causal Document Mask, Document Mask, and Shared Question Mask, were selected for evaluation.
>
> The datasets used were from Section A.5.2, but were slightly modified to ensure that each sub-document sequence length was divisible by 64, allowing for experiments with a FlashInfer sparse mask where C=64. As shown in the experimental results below, FlashMask demonstrated superior TFLOPs/s, effectively addressing the inefficiencies observed with FlashInfer's dense mask API. While FlashInfer with sparse masks shows considerable improvements as the mask block sizes (R and C) increase, especially when R, C ≥ 16, such large mask block sizes are rarely required in practice.
>
> Within FlashInfer’s `single_prefill_with_kv_cache` DenseMask implementation in [prefill.cuh#L1234-L1241](https://github.com/flashinfer-ai/flashinfer/blob/v0.1.6/include/flashinfer/attention/prefill.cuh#L1234-L1241), the use of token-by-token dense masks results in substantial inefficiencies, especially since calculations for fully masked blocks are entirely unnecessary. Regarding FlashInfer's `BlockSparseAttentionWrapper`, the mask block column $C$ functions as the page size. Smaller mask block sizes lead to a marked increase in the padded batch size `nblks(padded_batch_size, 1, num_kv_heads)`, which adversely affects performance because of non-optimal kernel hyper-parameter tuning. In contrast, with larger mask block sizes, FlashInfer maximizes the advantages of BSR's sparse representation by calculating only the necessary tiling blocks and avoiding excess calculations for fully masked blocks, thereby achieving high TFLOPs/s.
>
> The hyper-parameter tuning results of FlashInfer are as follows:
>
> ```
> R1C1: request_idx=8180, packed_qo_len=4, kv_len=2496, qo_chunk_size=16, kv_chunk_size=2496, num_tiles_q=1, num_tiles_kv=1
> R2C2: request_idx=4094, packed_qo_len=8, kv_len=960, qo_chunk_size=16, kv_chunk_size=1344, num_tiles_q=1, num_tiles_kv=1
> R4C4: request_idx=2047, packed_qo_len=16, kv_len=720, qo_chunk_size=16, kv_chunk_size=784, num_tiles_q=1, num_tiles_kv=1
> R8C8: request_idx=1023, packed_qo_len=32, kv_len=360, qo_chunk_size=64, kv_chunk_size=392, num_tiles_q=1, num_tiles_kv=1
> R16C16: request_idx=511, packed_qo_len=64, kv_len=180, qo_chunk_size=64, kv_chunk_size=196, num_tiles_q=1, num_tiles_kv=1
> R32C32: request_idx=255, packed_qo_len=128, kv_len=90, qo_chunk_size=128, kv_chunk_size=98, num_tiles_q=1, num_tiles_kv=1
> R64C64: request_idx=127, packed_qo_len=256, kv_len=45, qo_chunk_size=128, kv_chunk_size=49, num_tiles_q=2, num_tiles_kv=1
>
> R2C2: partition_kv=0, padded_batch_size=4096, num_warps_x=1, num_warps_z=4, num_frags_x=1, num_frags_y=8, num_frags_z=2
> R4C4: partition_kv=0, padded_batch_size=2048, num_warps_x=1, num_warps_z=4, num_frags_x=1, num_frags_y=8, num_frags_z=2
> R8C8: partition_kv=0, padded_batch_size=1024, num_warps_x=4, num_warps_z=1, num_frags_x=1, num_frags_y=8, num_frags_z=8
> R16C16: partition_kv=0, padded_batch_size=512, num_warps_x=4, num_warps_z=1, num_frags_x=1, num_frags_y=8, num_frags_z=8
> R32C32: partition_kv=0, padded_batch_size=256, num_warps_x=4, num_warps_z=1, num_frags_x=2, num_frags_y=8, num_frags_z=4
> R64C64: partition_kv=0, padded_batch_size=256, num_warps_x=4, num_warps_z=1, num_frags_x=2, num_frags_y=8, num_frags_z=4
> ```
>
> We hope this response clarifies our approach and experimental scope, and we look forward to any additional feedback you might provide.
>
> Best regards

---

> > ### Author Response · Authors · 2024-11-22
> >
> > **Comparison Results of FlashInfer DenseMask (single_prefill_with_kv_cache), FlashInfer SparseMask (BlockSparseAttentionWrapper), and FlashMask on Causal Document Mask at 8K, 32K, and 128K**
> >
> > | Method                | Mask Type            | Seq Length |   Sparsity |   FW Time (ms) |   FW TFLOPs |   FW TFLOPs/s |
> > |:----------------------|:---------------------|-----------:|-----------:|---------------:|------------:|--------------:|
> > | FlashInfer SparseMask | Causal Document Mask |       8192 |   0.880591 |       9.32757  |   0.131292  |      13.9464  |
> > | FlashInfer DenseMask  | Causal Document Mask |       8192 |   0.880591 |      11.9305   |   0.131292  |      11.0058  |
> > | FlashMask             | Causal Document Mask |       8192 |   0.880591 |       0.960801 |   0.131292  |     135.071   |
> > | FlashInfer SparseMask | Causal Document Mask |      32768 |   0.953204 |      54.7695   |   0.823251  |      15.0114  |
> > | FlashInfer DenseMask  | Causal Document Mask |      32768 |   0.953204 |     184.198    |   0.823251  |       4.46944 |
> > | FlashMask             | Causal Document Mask |      32768 |   0.953204 |       5.98586  |   0.823251  |     137.087   |
> > | FlashInfer SparseMask | Causal Document Mask |     131072 |   0.955792 |     788.939    |  12.4435    |      15.7707  |
> > | FlashInfer DenseMask  | Causal Document Mask |     131072 |   0.955792 |    2948.23     |  12.4435    |       4.22073 |
> > | FlashMask             | Causal Document Mask |     131072 |   0.955792 |      84.127    |  12.4435    |     147.582   |
> >
> >
> > **Comparison Results of FlashInfer DenseMask (single_prefill_with_kv_cache), FlashInfer SparseMask (BlockSparseAttentionWrapper), and FlashMask on Share Question Mask at 8K, 32K, and 128K**
> > | Method                | Mask Type            | Seq Length |   Sparsity |   FW Time (ms) |   FW TFLOPs |   FW TFLOPs/s |
> > |:----------------------|:---------------------|-----------:|-----------:|---------------:|------------:|--------------:|
> > | FlashInfer SparseMask | Share Question Mask  |       8192 |   0.932401 |       6.11861  |   0.0743262 |      11.7586  |
> > | FlashInfer DenseMask  | Share Question Mask  |       8192 |   0.932401 |      11.938    |   0.0743262 |       6.22726 |
> > | FlashMask             | Share Question Mask  |       8192 |   0.932401 |       0.727063 |   0.0743262 |      98.4887  |
> > | FlashInfer SparseMask | Share Question Mask  |      32768 |   0.974209 |      32.872    |   0.453728  |      13.7364  |
> > | FlashInfer DenseMask  | Share Question Mask  |      32768 |   0.974209 |     184.396    |   0.453728  |       2.46063 |
> > | FlashMask             | Share Question Mask  |      32768 |   0.974209 |       4.58727  |   0.453728  |      98.262   |
> > | FlashInfer SparseMask | Share Question Mask  |     131072 |   0.975079 |     443.804    |   7.01461   |      15.7953  |
> > | FlashInfer DenseMask  | Share Question Mask  |     131072 |   0.975079 |    2948.89     |   7.01461   |       2.37871 |
> > | FlashMask             | Share Question Mask  |     131072 |   0.975079 |      61.5699   |   7.01461   |     113.21    |

---

> ### Author Response · Authors · 2024-11-22
>
> **Comparison of FlashInfer DenseMask (single_prefill_with_kv_cache), FlashInfer SparseMask (BlockSparseAttentionWrapper), and FlashMask on Document Masks at an 8K Sequence Length, with Varying R/C**
>
> Although FlashInfer DenseMask and FlashMask do not have specific R/C values, the masks they represent are consistent under different R/C due to the total sequence length and each sub-document's sequence length being divisible by 64. We have included them in the table for comparison with FlashInfer SparseMask, conducting multiple tests to ensure a fair evaluation.
>
> | Method                | Mask Type            |   R/C | Seq Length |   Sparsity |   FW Time (ms) |   FW TFLOPs |   FW TFLOPs/s |
> |:----------------------|:---------------------|------:|-----------:|-----------:|---------------:|------------:|--------------:|
> | FlashInfer SparseMask | Document Mask        |     1 |       8192 |   0.786804 |      15.3937   |   0.234411  |      15.1884  |
> | FlashInfer SparseMask | Document Mask        |     2 |       8192 |   0.761304 |       8.56867  |   0.262449  |      30.482   |
> | FlashInfer SparseMask | Document Mask        |     4 |       8192 |   0.761304 |       4.31257  |   0.262449  |      60.5664  |
> | FlashInfer SparseMask | Document Mask        |     8 |       8192 |   0.761304 |       3.23164  |   0.262449  |      80.9699  |
> | FlashInfer SparseMask | Document Mask        |    16 |       8192 |   0.761304 |       1.64875  |   0.262449  |     158.545   |
> | FlashInfer SparseMask | Document Mask        |    32 |       8192 |   0.761304 |       1.51439  |   0.262449  |     172.609   |
> | FlashInfer SparseMask | Document Mask        |    64 |       8192 |   0.761304 |       1.5123   |   0.262449  |     172.817   |
> | FlashInfer DenseMask  | Document Mask        |     1 |       8192 |   0.786804 |      11.921    |   0.234411  |      19.6652  |
> | FlashInfer DenseMask  | Document Mask        |     2 |       8192 |   0.761304 |      11.911    |   0.262449  |      22.037   |
> | FlashInfer DenseMask  | Document Mask        |     4 |       8192 |   0.761304 |      11.9116   |   0.262449  |      22.0376  |
> | FlashInfer DenseMask  | Document Mask        |     8 |       8192 |   0.761304 |      11.9152   |   0.262449  |      22.0298  |
> | FlashInfer DenseMask  | Document Mask        |    16 |       8192 |   0.761304 |      11.9125   |   0.262449  |      22.035   |
> | FlashInfer DenseMask  | Document Mask        |    32 |       8192 |   0.761304 |      11.9135   |   0.262449  |      22.0331  |
> | FlashInfer DenseMask  | Document Mask        |    64 |       8192 |   0.761304 |      11.9119   |   0.262449  |      22.0358  |
> | FlashMask             | Document Mask        |     1 |       8192 |   0.786804 |       1.51063  |   0.234411  |     154.494   |
> | FlashMask             | Document Mask        |     2 |       8192 |   0.761304 |       1.68858  |   0.262449  |     154.789   |
> | FlashMask             | Document Mask        |     4 |       8192 |   0.761304 |       1.67081  |   0.262449  |     156.027   |
> | FlashMask             | Document Mask        |     8 |       8192 |   0.761304 |       1.66058  |   0.262449  |     156.736   |
> | FlashMask             | Document Mask        |    16 |       8192 |   0.761304 |       1.6599   |   0.262449  |     156.823   |
> | FlashMask             | Document Mask        |    32 |       8192 |   0.761304 |       1.65797  |   0.262449  |     157.007   |
> | FlashMask             | Document Mask        |    64 |       8192 |   0.761304 |       1.65883  |   0.262449  |     156.941   |

---

> ### Author Response · Authors · 2024-11-22
>
> **Comparison of FlashInfer DenseMask (single_prefill_with_kv_cache), FlashInfer SparseMask (BlockSparseAttentionWrapper), and FlashMask on Document Masks at an 32K Sequence Length, with Varying R/C**
>
> Although FlashInfer DenseMask and FlashMask do not have specific R/C values, the masks they represent are consistent under different R/C due to the total sequence length and each sub-document's sequence length being divisible by 64. We have included them in the table for comparison with FlashInfer SparseMask, conducting multiple tests to ensure a fair evaluation.
>
> | Method                | Mask Type            |   R/C | Seq Length |   Sparsity |   FW Time (ms) |   FW TFLOPs |   FW TFLOPs/s |
> |:----------------------|:---------------------|------:|-----------:|-----------:|---------------:|------------:|--------------:|
> | FlashInfer SparseMask | Document Mask        |     1 |      32768 |   0.906438 |     104.646    |   1.64597   |      15.7256  |
> | FlashInfer SparseMask | Document Mask        |     2 |      32768 |   0.906438 |      52.4631   |   1.64597   |      31.3644  |
> | FlashInfer SparseMask | Document Mask        |     4 |      32768 |   0.906438 |      25.9617   |   1.64597   |      63.47    |
> | FlashInfer SparseMask | Document Mask        |     8 |      32768 |   0.906438 |      19.6772   |   1.64597   |      83.5854  |
> | FlashInfer SparseMask | Document Mask        |    16 |      32768 |   0.906438 |       9.87305  |   1.64597   |     166.575   |
> | FlashInfer SparseMask | Document Mask        |    32 |      32768 |   0.906438 |       8.88797  |   1.64597   |     185.125   |
> | FlashInfer SparseMask | Document Mask        |    64 |      32768 |   0.906438 |       8.88604  |   1.64597   |     185.16    |
> | FlashInfer DenseMask  | Document Mask        |     1 |      32768 |   0.906438 |     184.097    |   1.64597   |       8.94137 |
> | FlashInfer DenseMask  | Document Mask        |     2 |      32768 |   0.906438 |     183.982    |   1.64597   |       8.94653 |
> | FlashInfer DenseMask  | Document Mask        |     4 |      32768 |   0.906438 |     183.995    |   1.64597   |       8.94587 |
> | FlashInfer DenseMask  | Document Mask        |     8 |      32768 |   0.906438 |     184.033    |   1.64597   |       8.94402 |
> | FlashInfer DenseMask  | Document Mask        |    16 |      32768 |   0.906438 |     183.995    |   1.64597   |       8.94599 |
> | FlashInfer DenseMask  | Document Mask        |    32 |      32768 |   0.906438 |     183.997    |   1.64597   |       8.94577 |
> | FlashInfer DenseMask  | Document Mask        |    64 |      32768 |   0.906438 |     183.986    |   1.64597   |       8.94627 |
> | FlashMask             | Document Mask        |     1 |      32768 |   0.906438 |      11.747    |   1.64597   |     139.665   |
> | FlashMask             | Document Mask        |     2 |      32768 |   0.906438 |      11.7424   |   1.64597   |     139.72    |
> | FlashMask             | Document Mask        |     4 |      32768 |   0.906438 |      11.7429   |   1.64597   |     139.715   |
> | FlashMask             | Document Mask        |     8 |      32768 |   0.906438 |      11.7343   |   1.64597   |     139.819   |
> | FlashMask             | Document Mask        |    16 |      32768 |   0.906438 |      11.7329   |   1.64597   |     139.836   |
> | FlashMask             | Document Mask        |    32 |      32768 |   0.906438 |      11.7327   |   1.64597   |     139.837   |
> | FlashMask             | Document Mask        |    64 |      32768 |   0.906438 |      11.7301   |   1.64597   |     139.869   |

---

> ### Author Response · Authors · 2024-11-22
>
> **Comparison of FlashInfer DenseMask (single_prefill_with_kv_cache), FlashInfer SparseMask (BlockSparseAttentionWrapper), and FlashMask on Document Masks at an 128K Sequence Length, with Varying R/C**
>
> Although FlashInfer DenseMask and FlashMask do not have specific R/C values, the masks they represent are consistent under different R/C due to the total sequence length and each sub-document's sequence length being divisible by 64. We have included them in the table for comparison with FlashInfer SparseMask, conducting multiple tests to ensure a fair evaluation.
>
> | Method                | Mask Type            |   R/C | Seq Length |   Sparsity |   FW Time (ms) |   FW TFLOPs |   FW TFLOPs/s |
> |:----------------------|:---------------------|------:|-----------:|-----------:|---------------:|------------:|--------------:|
> | FlashInfer SparseMask | Document Mask        |     1 |     131072 |   0.911591 |    1571.12     |  24.8848    |      15.8381  |
> | FlashInfer SparseMask | Document Mask        |     2 |     131072 |   0.911591 |     783.622    |  24.8848    |      31.7548  |
> | FlashInfer SparseMask | Document Mask        |     4 |     131072 |   0.911591 |     391.201    |  24.8848    |      63.6058  |
> | FlashInfer SparseMask | Document Mask        |     8 |     131072 |   0.911591 |     288.973    |  24.8848    |      86.1052  |
> | FlashInfer SparseMask | Document Mask        |    16 |     131072 |   0.911591 |     145.126    |  24.8848    |     171.447   |
> | FlashInfer SparseMask | Document Mask        |    32 |     131072 |   0.911591 |     131.312    |  24.8848    |     189.495   |
> | FlashInfer SparseMask | Document Mask        |    64 |     131072 |   0.911591 |     131.325    |  24.8848    |     189.476   |
> | FlashInfer DenseMask  | Document Mask        |     1 |     131072 |   0.911591 |    2946.7      |  24.8848    |       8.44507 |
> | FlashInfer DenseMask  | Document Mask        |     2 |     131072 |   0.911591 |    2946.88     |  24.8848    |       8.44471 |
> | FlashInfer DenseMask  | Document Mask        |     4 |     131072 |   0.911591 |    2947.32     |  24.8848    |       8.44338 |
> | FlashInfer DenseMask  | Document Mask        |     8 |     131072 |   0.911591 |    2947        |  24.8848    |       8.44419 |
> | FlashInfer DenseMask  | Document Mask        |    16 |     131072 |   0.911591 |    2946.96     |  24.8848    |       8.44435 |
> | FlashInfer DenseMask  | Document Mask        |    32 |     131072 |   0.911591 |    2946.73     |  24.8848    |       8.44511 |
> | FlashInfer DenseMask  | Document Mask        |    64 |     131072 |   0.911591 |    2946.81     |  24.8848    |       8.44475 |
> | FlashMask             | Document Mask        |     1 |     131072 |   0.911591 |     172.883    |  24.8848    |     143.616   |
> | FlashMask             | Document Mask        |     2 |     131072 |   0.911591 |     172.859    |  24.8848    |     143.635   |
> | FlashMask             | Document Mask        |     4 |     131072 |   0.911591 |     172.837    |  24.8848    |     143.654   |
> | FlashMask             | Document Mask        |     8 |     131072 |   0.911591 |     172.822    |  24.8848    |     143.666   |
> | FlashMask             | Document Mask        |    16 |     131072 |   0.911591 |     172.811    |  24.8848    |     143.675   |
> | FlashMask             | Document Mask        |    32 |     131072 |   0.911591 |     172.807    |  24.8848    |     143.679   |
> | FlashMask             | Document Mask        |    64 |     131072 |   0.911591 |     172.809    |  24.8848    |     143.677   |

---

> ### Author Response · Authors · 2024-11-22
>
> **Q: How can this technique be integrated with page attention?**
>
> A: FlashMask's innovative column-wise mask representation significantly reduces memory complexity by using an index range vector to characterize each column. It classifies tiling blocks into Fully masked, Partially masked, or Unmasked categories, allowing computations to be skipped for Fully masked blocks and optimizing other blocks by eliminating redundant mask applications. On the other hand, Page Attention deals with KV Cache tokens that are managed across discontinuous physical storage based on page size granularity. Since FlashMask reduces memory complexity from $O(N^2)$ to $O(N)$, the necessity for page management is eliminated. These two techniques are orthogonal, facilitating seamless integration. By leveraging FlashMask's capabilities, Page Attention can support a more sophisticated expression of masks.
>
> **Q: Can tree-based speculative decoding benefit from this customized attention?**
>
> A: Indeed, FlashMask supports tree-based speculative decoding methods, as depicted in [SpecInfer](https://arxiv.org/abs/2305.09781)'s Tree-based Parallel Decoding example (Figure 4), with sequences like $LTS=[1, 2, 3, 4, 5, 6, 7, 8]$, $LTE=[1, 2, 3, 4, 5, 6, 7, 8]$, $UTS=[0, 0, 0, 0, 3, 3, 2, 2]$, and $LTS=[0, 0, 0, 0, 4, 4, 6, 6]$.
>
>
> **Q: Can you report evaluation results on machines such as P100, V100, A10G, and H100? (other than A100)**
>
> A: The current version of FlashMask extends from FlashAttention-2 without substantial code restructuring, thus it is not supported on legacy architectures like P100 and V100. Unfortunately, due to the lack of H100 and A10G, we have yet to conduct evaluations on Hopper and other Ampere GPUs. However, the methodology itself is adaptable across hardware platforms.

---

> ### Author Response · Authors · 2024-12-03
>
> Dear Reviewer gr4R,
>
> I hope this message finds you well. I am writing to kindly follow up on my response to your feedback. I have addressed the questions and comments you raised, including conducting the additional experiments and providing the requested results. I sincerely appreciate the time and effort you have dedicated to reviewing my work, and I am grateful for your constructive feedback.
>
> If there is anything further you would like me to clarify or expand upon, please do not hesitate to let me know. Additionally, if the rebuttal has satisfactorily addressed your concerns, I would kindly appreciate your consideration in reflecting this in the final scoring.
>
> Thank you once again for your thoughtful feedback and your time. I look forward to your response at your earliest convenience.
>
> Best regards,
>
> The Authors

---

> > ### Comment · Reviewer_gr4R · 2024-12-03
> >
> > I think the experiments are convincing. I will raise my score.

---

> > > ### Author Response · Authors · 2024-12-03
> > >
> > > Thank you very much for your thoughtful feedback and for taking the time to review our rebuttal; we greatly appreciate your support and consideration. We will incorporate the experimental results into the revised version.

---

### Official Review · Reviewer_TH6Z · 2024-11-01

**Soundness:** 3
**Presentation:** 3
**Contribution:** 2
**Rating:** 6
**Confidence:** 5

**Summary:**

The paper proposes an efficient sparse mask representation by using composition of LT and RT range for expressing complex patterns. The proposed mask is compatible with FlashAttention-2 and can bring speed-up when applied.

**Strengths:**

1. The paper open-sourced a rather general sparse self-attention representation framework, which could facilitate many research and production attempts in the field.
2. The implementation is practical, shown wall-clock speed-up over FlashAttention-2.

**Weaknesses:**

1. It seems the implementation is limited to Paddle. It would be good to see if it can also be made more general so that the Torch/Megatron community can also leverage the framework.
2. Inference support is missing. It would make more sense to discuss how such sparse mask can be put into actual inference/serving.
3. [1] was published earlier, and also provide a general sparse self-attention training & serving framework. It would be ideal to also cite [1].

[1] S2-Attention: Hardware-Aware Context Sharding Among Attention Heads

**Questions:**

1. What would be the block size supported by Flashmask? Namely, what would be the granularity of mask/unmask chunks?
2. How does different block/chunk size affect the speed-up, in different mask types?
3. When tiling, it seems some mask may lead to different workload among thread blocks, which could hurt the overall performance. Is there any mitigation to this?
4. Can we have a comparison between the theoretical FLOPs reduction wrt wall-clock speed-up for different mask types?
5. How does tensor parallel and pipeline parallel affect the speed-up?

---

> ### Author Response · Authors · 2024-11-22
>
> Dear Reviewer,
>
> Thank you for the valuable feedback and questions on our paper. We appreciate the chance to elaborate on our work and address your concerns.
>
> **Weaknesses 1: Implementation limited to PaddlePaddle.**
>
> Our current implementation of FlashMask is indeed conducted within the PaddlePaddle framework. However, it is integrated as a third-party module in PaddlePaddle and decouples from framework, so it's easy to be integrated into PyTorch or other frameworks. We aim to expand our validation efforts to ensure broader compatibility and adoption across platforms.
>
> **Weaknesses 2: Inference support is missing.**
>
> FlashMask can certainly be extended to inference applications, like Page Attention and tree-based speculative decoding. Although this paper primarily focuses on training aspects, we acknowledge the importance of discussing inference use cases and plan to explore these in future work.
>
> **Weaknesses 3: Missing citation to [1].**
>
> We appreciate your suggestion regarding [1] (S2-Attention: Hardware-Aware Context Sharding Among Attention Heads). It is indeed a commendable work, and we intended to cite it in the camera-ready version of our submission. S2-Attention employs a Block-Sparse Attention strategy focusing on approximate attention sparsity calculations at both per-head and per-context-range levels for long-sequence modeling. In contrast, FlashMask emphasizes supporting a wide variety of attention masks used in transformer training through precise computation rather than approximate methods. Our approach reduces memory complexity to $O(N)$ by introducing a novel column-wise mask representation. This enables handling longer sequences efficiently while supporting mainstream attention masks precisely.
>
> Thank you once again for your thoughtful review and questions, and we look forward to refining FlashMask further.
>
> Best regards

---

> ### Author Response · Authors · 2024-11-22
>
> **Q: What would be the block size supported by FlashMask? Namely, what would be the granularity of mask/unmask chunks?**
>
> A: In response to your question, we define the "block size" or "mask/unmask chunks" as "mask block size." FlashMask represents masks with token-level granularity, thereby setting the mask block size to 1. This minimal granularity supports arbitrary mask block size. As detailed in sections 4.1 and 4.2, FlashMask divides tiling blocks into Fully masked, Partially masked, and Unmasked categories by representing each column with a range of indices, thereby bypassing computation on Fully masked blocks while applying masks only to Partially masked blocks.
>
> **Q: How does different block/chunk size affect the speed-up, in different mask types?**
>
> A: Again, defining "block size" or "chunk size" as "mask block size" and differentiating from the computation tiling block size in FlashAttention Kernels, FlashMask maintains a mask block size of 1. Thus, it supports any mask block size. The tiling block size remains unchanged from FlashAttention-2. FlashMask primarily focuses on reducing memory complexity via the novel column-wise mask representation. Our experiments (see Section 5.3, Figure 4a) reveal a strong correlation between block sparsity and processing latency—more structured attention masks result in fewer partially masked blocks, thereby decreasing the computational load and enhancing performance. It should be noted that FlashMask is not yet optimized for maximum performance, as there are additional memory accesses and condition checks that have not been fully optimized. In the future, we plan to further optimize these aspects.
>
> **Q: When tiling, it seems some mask may lead to different workload among thread blocks, which could hurt the overall performance. Is there any mitigation to this?**
>
> A: The current FlashMask implementation inherits from FlashAttention 2, assigning each thread block to handle calculations for a given batch, head, and tiling row. Despite potential workload imbalances due to varying mask types within tiling blocks (Fully masked, Partially masked, Unmasked), the SMs (Streaming Multiprocessors) remain fully utilized as multiple thread blocks can be assigned to each SM. Thus, mask diversity does not negatively impact overall performance. That said, future optimizations will focus on reducing unnecessary memory access and conditions, as depicted in Algorithm 1, lines 9-14.
>
> **Q: Can we have a comparison between the theoretical FLOPs reduction and wall-clock speed-up for different mask types?**
>
> A: As stated in Section 4.3, the theoretical FLOPs reduction is linearly related to the block sparsity ($\rho$) in the attention mask and is independent of the specific mask type. This indicates that kernel latency should be proportional to $\mathcal{O}((1-\rho)T_rT_c)$. The actual speed-up observed in the experiments detailed in Figure 4(a) further supports this relationship, showing a strong correlation between the theoretical and empirical results.
>
> **Q: How does tensor parallelism and pipeline parallelism affect the speed-up?**
>
> A: FlashMask is an optimization at the operator level, and its effects are orthogonal to those of tensor parallelism and pipeline parallelism. These parallel strategies distribute computation equally across different devices. Therefore, we believe that different parallel strategies have minimal impact on the performance gains provided by FlashMask.

---

> ### Author Response · Authors · 2024-12-03
>
> Dear TH6Z,
>
> I hope you’re doing well. I wanted to follow up on my rebuttal, where I’ve addressed your questions and comments. I sincerely appreciate your time and constructive feedback.
>
> If there’s anything else you’d like me to clarify or expand on, please let me know. Your insights are invaluable in improving my submission.
>
> Thank you again for your thoughtful review. I look forward to your response.
>
> Best regards,
>
> The Authors

---

### Official Review · Reviewer_Uyji · 2024-11-03

**Soundness:** 3
**Presentation:** 4
**Contribution:** 3
**Rating:** 6
**Confidence:** 4

**Summary:**

The paper introduces a novel compression scheme for the attention mask where only the boundary indices of the masks are stored for every column. For a specific set of attention masks, it is sufficient to store two sets of boundary indices for every column to represent the attention mask. This reduces the memory complexity of attention masks from quadratic on sequence length to linear on sequence length, enabling handling of longer context lengths. The column-wise sparse representation is also used to skip fully masked blocks increasing the overall compute efficiency of the attention mechanism. This technique is augmented with FlashAttention algorithm for efficient computation of the attention mechanism and the modified algorithm for both forward pass and backward pass are presented. The experiments section shows that FlashMask is faster than FlashAttention dense method by up to 3.22x and can achieve up to 60.23% more throughput than FlexAttention. The proposed method also doesn’t alter the convergence during training.

**Strengths:**

The paper is well-written and easy to understand. The results section is elaborate with a wide range of benchmarks to demonstrate the advantages of the proposed methods. The appendix section and the analysis with synthetic data to corroborate the claims are very insightful. The compute and memory utilization advantages of FlashMask are well demonstrated. The proposed sparse representation scheme is novel and should be adopted wherever applicable for its memory efficiency and ability to support longer context lengths.

**Weaknesses:**

While the results section shows that FlashMask achieves higher computational efficiency, I’m not sure if it’s attributable to the proposed columns-wise sparse representation.
The computational efficiency of FlashMask comes from skipping computation on entirely masked blocks as discussed in section 4.3. However, this technique is also used in Block-Sparse FlashAttention and FlexAttention. The advantages of FlashMask over Block-Sparse FlashAttention and FlexAttention in terms of computational efficiency is not clear.
Also as mentioned in the paper, the idea of column-wise sparse representation used in FlashMask is limited to specific attention patterns. Any other pattern can’t be handled even naively.

**Questions:**

You mentioned FlexAttention can also exploit sparsity by skipping computation on fully masked blocks. If that’s the case where’s compute throughput advantage of FlashMask coming from?
Would the Block-Sparse FlashAttention be able to handle the mask types described in Fig 1(a)? If yes, that should be used instead of the DenseMask variant for the throughput comparisons across the paper. If not, please mention why.
In Fig 4b, why is the FlexAttention’s memory utilization lower than that of FlashMask for sequence length lower than 16K?

---

> ### Author Response · Authors · 2024-11-22
>
> Dear Reviewer,
>
> Thank you for your insightful feedback and questions regarding our paper. We appreciate the opportunity to address your concerns and provide additional clarifications.
>
> **Q: You mentioned FlexAttention can also exploit sparsity by skipping computation on fully masked blocks. If that’s the case where’s compute throughput advantage of FlashMask coming from?**
>
> A: Similar to FlexAttention, FlashMask also capitalizes on the sparsity of attention masks. The key distinction is that FlashMask is built upon FlashAttention2, inheriting all its manual and hyperparameter optimizations. As demonstrated in the performance chart on the official FlexAttention blog, the current FlexAttention is slower than FlashAttention2. For more details, you can refer to the blog post at [https://pytorch.org/blog/flexattention/#performance](https://pytorch.org/blog/flexattention/#performance). FlexAttention, however, is still in development, utilizing Triton-based compiler technology, and has not yet reached its full performance potential.
>
> Moreover, although FlashMask's current performance is not optimal, as it only achieves 37.8% to 62.3% of the theoretical maximum FLOPs/s on A100 GPUs, we plan to enhance it by reconstructing the traversal method, further improving throughput for all mask types.
>
> **Q: Would the Block-Sparse FlashAttention handle the mask types described in Fig 1(a)? If yes, it should be used instead of the DenseMask variant for throughput comparisons across the paper. If not, please mention why.**
>
> A: No. As explained in Section 2.3, Block-Sparse FlashAttention represents masks with tiling-level granularity, which makes it unsuitable for handling token-level masks like DenseMask. Previously, DenseMask was the only comprehensive method capable of representing the attention masks shown in Fig 1(a). Therefore, we used DenseMask as a performance baseline in our paper.
>
> Furthermore, Block-Sparse FlashAttention was developed as part of a new approximate attention algorithm. In contrast, our FlashMask is designed to accelerate the attention module when the sparsity of the attention mask naturally arises from the problem itself. For instance, the causal document mask, which is inherently sparse, is commonly used in SFT training. Unlike the approximate computations in Block-Sparse FlashAttention, our approach leverages the inherent sparsity of the mask to achieve acceleration without any loss of precision.
>
>
> **Q: Why is the FlexAttention’s memory utilization lower than that of FlashMask for sequence length lower than 16K in Fig 4b?**
>
> A: As discussed in section 2.2, the memory complexity for FlexAttention is $O\left(\frac{N^2}{{B_c}{B_r}}\right)$, where $B_r=128$ and $B_c=128$. FlashMask, on the other hand, employs an innovative column-wise mask representation, maintaining a linear memory complexity of $O(N)$. When the sequence length is less than 16K, $O\left(\frac{N^2}{{B_c}{B_r}}\right)$ is actually less than $O(N)$. Although the FlexAttention blog suggests potential memory savings by increasing the block sizes $B_r$ and $B_c$, its complexity still remains quadratic with respect to the sequence length $N$. Conversely, when the sequence length exceeds 16K, FlashMask's memory usage is lower than that of FlexAttention.
>
>
> We appreciate your thorough scrutiny and hope these responses satisfactorily address your concerns.
>
> Best regards

---

> ### Author Response · Authors · 2024-12-03
>
> Dear Uyji,
>
> I hope you’re doing well. I wanted to follow up on my rebuttal, where I’ve addressed your questions and comments. I sincerely appreciate your time and constructive feedback.
>
> If there’s anything else you’d like me to clarify or expand on, please let me know. Your insights are invaluable in improving my submission.
>
> Thank you again for your thoughtful review. I look forward to your response.
>
> Best regards,
>
> The Authors

---

### Meta-Review · Area_Chair_f9As · 2024-12-15

**Metareview:**

The paper presents further optimization over Flash-Attention 2 for better memory management of the Attention mechanism, I think the novelty of the approach in this paper is limited. But given the reviewers are enthusiastic about it, it would be ok to accept.

**Additional Comments On Reviewer Discussion:**

Rebuttal led to increase of score.

---

### Decision · Program_Chairs · 2025-01-22

Accept (Poster)